# Testing the oceanic dispersal potential of Caribbean fleshy-fruited plants

Seokmin Kim[1]*, Sofany Montoya[1], Fabio L. Tarazona-Tubens[1], Christina Chavez[2], Joanna Tucker Lima[2], Donald Olson[3], Christopher Searcy[1]*

1 Department of Biology, University of Miami, Coral Gables, Florida, United States of America,
2 Montgomery Botanical Center, Coral Gables, Florida, United States of America, 3 Department of Ocean Sciences, University of Miami, Coral Gables, Florida, United States of America

* kimseokmin2401@gmail.com (SK); christopher.searcy82@gmail.com (CS)

## Abstract

Thalassochory, or dispersal by ocean currents, shapes island biogeographical processes. However, the potential of fleshy-fruited plants to utilize this dispersal method is understudied. We selected 14 fleshy-fruited species found in coastal Caribbean plant communities and assessed their thalassochoric dispersal potential by measuring the period during which they could both float and remain viable in saltwater. We then determined the thalassochoric connectivity between 44 Caribbean islands by analyzing the paths of 1198 drifter buoys that passed through the Caribbean between 1991 and 2019. We found a significant, positive trend for fruits with greater thalassochoric dispersal potential to be found on more islands and significant variation in the floating potential of Caribbean fleshy-fruited plants (0–90 + days), with the species with the greatest floating potential (*Chrysobalanus icaco*) feasibly being able to disperse viable seeds between the most geographically distant (2600 km) pair of islands in the Caribbean. However, we could not tie thalassochoric connectivity to either individual species distributions or community composition across our 14 fleshy-fruited species. Geographic distance, the isolation metric traditionally used in island biogeography studies, could not explain distribution or community patterns either, while human usage was identified as an alternative significant predictor of species range size. Our results thus illustrate the challenges associated with identifying drivers of distribution patterns across the Caribbean archipelagos. While we assert that future studies, such as those examining genetic connectivity of plant populations in the context of thalassochoric connectivity, are needed, this study serves as a crucial first step in understanding the role of sea currents in the distribution of fleshy-fruited plants.

**Data availability statement:** All data used in this paper, relevant supporting materials, and supplemental tables are available through GitHub (https://github.com/sxk1332/floating_fruits).

**Funding:** The author(s) received no specific funding for this work.

**Competing interests:** The authors have declared that no competing interests exist.

## Introduction

Dispersal and connectivity are key aspects of understanding patterns of biodiversity due to the link they provide between regional and local populations [1]. These processes directly shape gene flow across a broad, regional space [2] and have implications for both conservation management and biogeography [3,4]. Long-distance dispersal (LDD), infrequent dispersal events across large geographic distances, allows genetic connectivity between distant populations [5] and is particularly crucial for the colonization of isolated locations [6]. Islands are ideal study systems for investigating LDD due to the clear delineation of species pools and the presence of clear dispersal barriers [7]. Islands are also experiencing high rates of species decline and extinction, particularly among large-bodied vertebrates [8]. This may impact patterns of plant LDD if the lost vertebrates were important vectors for seed dispersal.

Endozoochorous, fleshy fruited plants are an interesting group to study in this context, given their strong dependence on frugivorous animals for dispersal. Endozoochoric dispersal relies heavily on 'matching' functional traits between the interacting partners such as fruit size and frugivore gape size [9], as this determines the range of fruit sizes that a frugivore can consume and disperse [10]. This highlights the problem confronted by large-fruited plants when large-bodied vertebrates go extinct, as has been happening extensively in island systems [11,12]. When such defaunation occurs, large-fruited plants can lose the dispersal vector that they have co-evolved with [13,14], leading to a decoupling between the plant's evolved dispersal syndrome and its dispersal vector.

Another mechanism potentially involved in LDD of fleshy fruited plants is thalassochory, or dispersal by ocean currents [15,16]. Thalassochory is one of the main long-distance dispersal mechanisms for colonization of oceanic islands and thus plays a crucial role in our understanding of island biogeography [17]. Thalassochory shapes population structure by connecting populations separated by great oceanic distances [18,19] and has been shown to predict distributions and genetic connectivity of both marine and terrestrial organisms. For instance, Galindo et al. [20] showed that Caribbean coral reefs that are better connected by ocean currents tend to be more genetically similar, and Cowen et al. [21,22] demonstrated that Caribbean reef fish species can disperse up to 100 km using ocean currents, thus promoting the connectivity of fish populations across a broad regional area. Similar processes also occur among terrestrial organisms, with Arjona et al. [23] finding that genetic differences between populations of a coastal shrub species in the Galápagos Islands (*Cryptocarpus pyriformis*) are best explained by ocean current connectivity, and Miryeganeh et al. [24] highlighting the potential for distant populations of water-dispersed *Ipomoea pes-caprae* to be maintained by ocean-current mediated long-distance seed migrations. Additionally, ocean currents have been shown to facilitate the movement of species not specialized for water dispersal, with Calsbeek and Smith [25] showing that over-water dispersal of lizards (*Anolis* spp.) after storms facilitates gene flow between islands in the Bahamas.

There are other mechanisms by which plants can disperse between islands such as anemochory – wind dispersal [26] – and epizoochory – attachment to feathers,

fur, or skin [27]. However, LDD of large-fruited plants, notably those with fruits that are too large to be consumed and dispersed by persisting animals, may depend more on thalassochory. A classic example of such species is the coconut (*Cocos nucifera*), which almost exclusively depends on ocean currents for natural dispersal and has evolved traits to facilitate this process [28]. Successful thalassochory for fruited plants requires the propagule (fruit or seed) to float and the seed to stay viable while at sea – both of which have been documented for plants in coastal regions and islands [29,30].

In the Caribbean archipelagos, there is wide variation in the range size of large, fleshy-fruited plants, with species such as *Chrysobalanus icaco* being widely distributed throughout the region, while other species such as *Syagrus amara* and *Acrocomia crispa* have limited distributions [31]. While it is possible that these distributions may be a product of anthropogenic introductions [32] or differing range sizes of their historical frugivorous partners [33], such discrepancies could also be due to some plants showing greater capacity to engage in non-biotic forms of dispersal such as thalassochory. This is particularly important to consider in light of the decline of large-bodied frugivorous species in the Caribbean [34], which will make non-biotic forms of dispersal even more important for maintaining plant ranges in the future.

Here, we aimed to model thalassochoric dispersal potential for fleshy-fruited plant species native to the Caribbean archipelagos, and to assess how well their dispersal potential predicts current distributions. We address the following: (1) Do the fruits of some plant species float better than others? (2) Can fruit traits (size/density) predict floating potential? (3) Does dispersal potential by ocean currents predict fleshy-fruited plant species distributions? (4) Can human usage serve as an alternative predictor of species distributions? We expect to find significant variation in species' ability to float in ocean currents, with larger, less dense fruits benefiting most from thalassochory. We also predict that plants with fruits that float better have wider distributions than those that do not, and that connectivity metrics based on ocean currents will provide a better predictor of species distributions than simple geographic distance.

## Methods

### Study system

The Caribbean archipelagos have been identified as a biodiversity hotspot [35], possessing elevated levels of endemism yet suffering from high levels of habitat loss [36]. For our study, we considered 44 major (here considered as those larger than 100 km$^2$) islands within this region. Oceanographically, the region is dominated by the Caribbean Current, which is mainly formed by the conjunction of the eddy-dominated Guyana Current along the South American coast, the North Equatorial Current from the mid-Atlantic, and the Antilles Current that flows westwards along the northern Caribbean. The last two are part of the North Atlantic subtropical gyre, while the first connects the Caribbean to Brazil and southern hemisphere waters. The Caribbean Current flows westward from the Lesser Antilles towards the Yucatán peninsula before turning northward as the Yucatán Current and then eventually becoming the Gulf Stream heading northeast. These main flows are accompanied by robust eddy fields, which can widely and variably disperse propagules (Fig 1; [37,38]). Biologically, this circulation has played a crucial role in linking populations across seascapes [20,21,39].

### Fruit floating viability

We focused on species native to islands around the Caribbean with fruits of a wide variety of sizes (Table 1). We selected species that occur in the lowlands of each island, where they are most likely to benefit from ocean current dispersal, and obtained specimens from the Montgomery Botanical Center and Fairchild Tropical Botanical Garden in Coral Gables, Florida, USA. We relied on an existing dataset [40] for information regarding the size and mass of each species' fruit (defined as the entire fruiting body). For distribution, we determined the presence/absence of each species on each of our 44 study islands using verified records from the Global Biodiversity Information Facility (GBIF) through the *spocc* package in R [41,42].

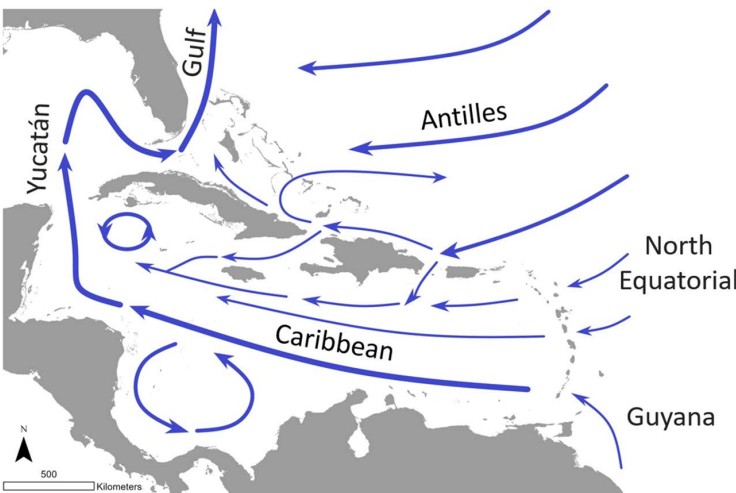

**Fig 1. Study region.** Ocean currents within our study region. Major currents are labeled.

**Table 1. Study species.**

| Species | Family | mean diam. (mm)* | mean length (mm)* | mean mass (g)* | total count | Max. floating day | Max. viable day | 95th percentile viable day | No. of islands recorded |
|---|---|---|---|---|---|---|---|---|---|
| *Acrocomia crispa* | Arecaceae | 23.9 | 24.3 | 3.9 | 111 | 21 | 21 | 21 | 2 |
| *Aiphanes minima* | Arecaceae | 15.7 | 16.1 | 2.7 | 63 | 13 | 13 | 9 | 8 |
| *Coccothrinax barbadensis* | Arecaceae | 10.9 | 12.4 | 1.1 | 168 | 0 | 0 | 0 | 12 |
| *Coccothrinax borhidiana* | Arecaceae | 12.9 | 15.2 | 1.8 | 182 | 3 | 3 | 3 | 1 |
| *Coccothrinax spissa* | Arecaceae | 10.4 | 12.0 | 1.1 | 77 | 0 | 0 | 0 | 1 |
| *Gaussia attenuata* | Arecaceae | 11.3 | 16.1 | 1.1 | 340 | 0 | 0 | 0 | 2 |
| *Pseudophoenix vinifera* | Arecaceae | 19.0 | 28.6 | 5.0 | 152 | 3 | 3 | 3 | 1 |
| *Syagrus amara* | Arecaceae | 31.0 | 41.0 | 25.2 | 29 | 0 | 0 | 0 | 4 |
| *Thrinax radiata* | Arecaceae | 8.8 | 10.3 | 0.6 | 168 | 28 | 28 | 11 | 15 |
| *Zombia antillarum* | Arecaceae | 20.5 | 21.4 | 5.2 | 33 | 21 | 21 | 21 | 1 |
| *Chrysobalanus icaco* | Chrysobalanaceae | 25.9 | 28.2 | 9.5 | 76 | 90+ | 90 | 87 | 33 |
| *Theophrasta jussieui* | Primulaceae | 43.4 | 46.7 | 49.0 | 17 | 28 | 21 | 7 | 1 |
| *Catesbaea spinosa* | Rubiaceae | 27.5 | 34.1 | 12.5 | 60 | 21 | 21 | 21 | 10 |
| *Goetzea elegans* | Solanaceae | 18.8 | 22.6 | 3.9 | 90 | 7 | 7 | 7 | 2 |

**Table 1**. Physical traits, total fruit count used in the study, floating potential, and range sizes for the 14 study species. *: trait information provided by an unpublished database [40].

Determination of the floating potential of fruits in salt water involved two methods. In the first method, we followed previous studies [30] and placed fruits from each species in a 42.5 x 29.5 x 17.8 cm container of salt water (S1 Fig). We then recorded the number of floating fruits each day for 100 days. Salt water was created from distilled water and Instant Ocean™ in concentration with a specific gravity range of 1.02–1.024 at room temperature. Each day, the contents of the container were stirred vigorously with a rod for 10 seconds to simulate water movement, and every fruit found to have sunk was removed. At the end of each week, we carefully replaced about half of the water to avoid eutrophication.

In the second method, we placed fruits from each species within a body of free moving salt water (mangrove forest edge located within Montgomery Botanical Center). In addition to providing circulating salt water, this location also gave protection (through the presence of mangroves) from any major wave action that might have disturbed our study. Devices to protect the floating fruits were constructed by securing fine mesh around a ring of foam (pool floaties) and attaching the bottom to an anchor (S2 Fig). The mesh protected the fruits from frugivores, while the foam prevented the mesh from weighing the fruits down, and the anchor prevented the fruits from floating away. The floating status of fruits placed within these devices was monitored twice a week for 90 days. This method more closely imitates natural conditions and avoids the issue of water eutrophication by providing constant flowing water. However, its environmental conditions such as salinity, temperature, and turbulence are difficult to control. Furthermore, we were unable to check for floatability every day due to logistical constraints. While we were able to conduct the second method for all species, due to limited specimen availability, we were only able to conduct the first method for five species. As we did not detect a difference in viability between the two methods through a paired t-test (t = 0.05, df = 4, p = 0.96), we combined our data from both methods for our analyses.

Using the fruits that we extracted from our floating apparatuses, we assessed the viability of intact seeds through the Tetrazolium (2,3,5-Triphenylte-trazolium chloride) test [43,44]. This method detects the viability of seeds by assessing the seeds' dehydrogenase enzymes, which catalyze mitochondrial respiration. During respiration, hydrogen ions are released, which react with tetrazolium to create formazan, a reddish substance. To conduct the tetrazolium test, we arbitrarily removed 10% of the floating fruits per species from the floating apparatuses once a week and extracted the seeds. If there were fewer than five fruits remaining for a given species, we did not remove any fruits for two more weeks and then checked all remaining fruits for viability. After washing the extracted seeds in distilled water, we made a longitudinal cut across the seed, taking care not to harm the embryo. These seed halves were then immersed in 1% tetrazolium and placed within a 35°C oven for 240 minutes, as detailed in Lautenschlager et al. [45]. The products were visually examined for clear, red markings to determine the viability of the seed (S3 Fig). We assumed that the proportion of viable seeds in our floating apparatuses were the same as the proportion of seeds shown to be viable through the tetrazolium test.

From these tests, we calculated both the maximum and 95th percentile viability times for each species (Table 1). We then used these metrics to assess relationships with volume and density of fleshy-fruited species (Table 1). We considered each fruit as an ellipsoid and used fruit width and length to calculate its volume. We then divided mass by volume of the entire fruit to calculate its density. This involved consulting an existing database of fruit traits [40], which used digital calipers and scale to measure relevant traits of at least ten specimens from each of 263 fleshy-fruited plant species native to the Caribbean archipelagos that were obtained from Montgomery Botanical Center, the Gifford Arboretum at the University of Miami, Fairchild Tropical Botanical Garden, Biscayne National Park, and field collections in several Caribbean islands (Virgin Islands, Lesser Antilles, Bahamas, etc.). We then conducted generalized linear mixed model regressions fitted with a Poisson family with one of the two viability metrics (maximum or 95th percentile) as the response variable and fruit volume or density as the independent variable, with plant family as a random effect through the *lme4* R package [46]. To assess the difference in viability potential between the study species, we used a modified survival analysis approach through the *survival* R package [47]. This method is commonly used in public health to assess and compare the effectiveness of specific traits or treatments on patient survival rates [48]. We modified this approach to evaluate viability times and considered the time it took for each fruit to either sink or cease to be viable.

## Dispersal potential and connectivity

Population connectivity studies in the Caribbean and elsewhere have largely been carried out using large scale physical oceanographic models to provide ocean current data from which trajectories are then calculated [21,22,49]. However, ocean drifter records provide another avenue of approach, which could provide more real-time, accurate results. Olson [50] conducted comparisons between model derived trajectories and ocean drifter records in the Caribbean and found

them to be consistent. At the time, however, they found that the existing drifter dataset did not consist of a large enough number of trajectories to provide full connectivity estimates. Given the many subsequent drifter deployments as part of the operational extension of the World Ocean Circulation Experiment (WOCE) drifter program [38,51,52], we believe that there are enough drifters available now to conduct full connectivity estimates, and this study represents the first to our knowledge to consider this approach for modeling thalassochoric connectivity in the Caribbean.

We obtained our drifter data from the Global Drifter Program (GDP), a principal component of the Global Surface Drifting Buoy Array, which is coordinated by the National Oceanic and Atmospheric Administration (NOAA). This data receives relevant information (position, velocity, etc.) from all registered drifters and interpolates them to six-hour intervals through kriging [52]. Our study area was between 10.3 and 27.4 latitude and −86.3 and −58.8 longitude, which encompasses all 44 islands within our study. A total of 1198 drifter buoys (from January 30, 1991 to November 4, 2019) entered this study region from all sides, 832 of which passed within half a degree (~50 km) of the study islands, providing data on ocean current linkages between island pairs. On average, each drifter moved between the islands for a period of 75 days before exiting the region (S4 Fig).

Our strategy for modelling connectivity and dispersal depends on matching surface drifter buoy IDs across various islands, allowing for the tracking of individual buoy movements and estimation of inter-island transit times. This involves (1) selecting a series of points of interest around the coastline of every study island (S5 Fig), (2) selecting a search area around each of the points, (3) determining how many shared buoys were found within the search areas around each island pair, (4) calculating the proportion of buoys that make it from one island to another, and (5) estimating the transit time between each pair of islands.

Selecting a search area is dependent on local water turbulence and therefore varies for each point. The final search area around each point that was used to compute arrival estimates is based on the mean flow ($< \overline{u} >$) and on the time scale of local currents ($< \overline{T} >$; [36]). Each parameter is calculated using characteristics of every recorded buoy within an initial, standardized search area around each point (0.5 square degrees for this study, n = 217), with the time scale calculated by the mean retention time (i.e., how long it takes for each buoy to pass through the initial search area) and the mean flow of the eastward and northward velocities. These parameters are then multiplied ($L_{xy} = < \overline{u} > * < \overline{T} >$) for each direction to determine the final search area around each of the points in our study area.

To model the degree of connectivity for each island in the study area, we pooled every point of interest and their associated search areas for each of the 44 islands in the study region. We then determined every buoy present in each island's pooled search area. Using this information, we first assessed the particle dispersal probability between each pair of islands by calculating how many buoys from a source island make it to a destination island through the following steps: (1) counting how many buoys were present around each source island, (2) selecting shared buoys between each pair of islands, (3) isolating buoys in the destination island that have time stamps later than those in the source island, and (4) dividing the number of shared buoys with the appropriate time stamps by the total number of buoys visiting the source island. Second, we determined the minimum dispersal time between each pair of islands by examining the time differences for the shared buoys. By pooling the transit times for every shared buoy, we developed a distribution (S6 Fig) from which the minimum value can be extracted. Following this framework, we created two pair-wise matrices for the study region that estimate: (1) the particle dispersal probability between each pair of islands (henceforth "dispersal probability matrix"; S1 Table) and (2) the minimum dispersal time (henceforth "dispersal time matrix"; S2 Table).

To calculate the level of thalassochoric connectivity in the region for every study species (S3 Table), we first evaluated the survival curve of each species. We then used this information in the context of the transit times for each shared buoy between islands and their dispersal probability (*P*) to estimate the proportion of fruits that could successfully move between each island pair, as described in the following equation:

$$Connectivity = \sum_{1}^{n} P(Plant\ survival\ |Time_n) * P(Dispersal\ |Time_n)$$

where "Plant survival" is the percentage of viable floating seeds, "Dispersal" is the fraction of buoys from the source island that reaches the destination island, and "$Time_n$" is the number of days ranging from 1–90 days (the time scale of our study).

### Statistical analyses

To assess correlations between geographic distance and thalassochoric dispersal potential, we first created a pair-wise matrix that estimated the shortest geographic distance between each pair of study islands using the *geosphere* package in R [52]. We then conducted Spearman's Mantel tests between geographic distance and the two regional thalassochoric dispersal metrics (dispersal probability matrix and dispersal time matrix). To determine if one of the two thalassochoric dispersal metrics was more correlated with geographic distance, we calculated Meng, Rosenthal, and Rubin's z statistic [53] through the *cocor* package in R [54]. We then carried out three tests to assess if thalassochoric dispersal potential is predictive of species distributions. First, we conducted a generalized linear mixed model using a Poisson error distribution, through the *lme4* package in R [46], with species viability (maximum or 95th percentile) as the predictor variable (square root-transformed to increase normality) and species range size (number of islands) as the response variable, with plant family as a random effect. To account for the potential for species distributions being shaped by humans, we also summarized each plant's potential for human use by looking up its genus in Molina-Venegas et al. [55], recording the number of human usage categories for that genus, and incorporating this human use metric into the generalized linear mixed model as an additional predictor variable. Second, we assessed correlations between each species' connectivity matrix and each species' distribution matrix using a Spearman's Mantel test for the ten species that had greater than zero maximum viability times. For the distribution matrices (S4 Table), we considered joint presences as 1, joint absences as NA, and differing absences/presences as 0. Third, we created two community similarity matrices using the Sørensen similarity index for species that are either good floaters (S5 Table) or poor floaters (S6 Table) based on whether they can float for more than the median 95th percentile viability time of all study plants (7 days), and assessed correlation between these matrices and the thalassochoric dispersal metrics (dispersal probability matrix and dispersal time matrix) using Spearman's Mantel test. We repeated the second and third steps with geographic distance between island pairs instead of connectivity metrics to assess any correlations between geographic distance and species distributions.

### Results

We used 14 fleshy-fruited species for this study. The largest species in our study was *Theophrasta jussieui* (20.5 mm diameter) and the smallest was *Thrinax radiata* (8.8 mm), the densest was *Coccothrinax spissa* (1.56 g/cm$^3$) and the least dense was *Acrocomia crispa* (0.54 g/cm$^3$). *Chrysobalanus icaco* was the most widely distributed species, present in 33 out of 44 islands in our study area. Meanwhile, five species (*Coccothrinax borhidiana, Coccothrinax spissa, Pseudophoenix vinifera, Theophrasta jussieui, Zombia antillarum*) were only recorded on one island (Table 1).

Our survival analysis found significant variation in floating potential between study species (Fig 2; $\chi^2 = 1215$, df = 13, $p < 0.001$). Four species (*Coccothrinax barbadensis, Coccothrinax spissa, Gaussia attenuata, Syagrus amara*) either sank immediately or were able to float for less than a day. In contrast, the species with the greatest floating/viability potential was *C. icaco*, in which 6% of the fruits stayed floating and viable until the end of our study at 90 days. Most of the species remained viable until they sank. The only exception was *T. jussieui*, in which the last individual sank after 28 days and the last time seeds were shown to be viable was after 21 days. There was no relationship between volume or density of fruits and their ability to float/stay viable (volume: t = 0.57, df = 11.9, p = 0.58; density: t = −1.23, df = 9.13, p = 0.25).

Of the 44 islands used in the study, seven of the islands (Gonave, Tortuga, Providenciales, South Andros, Mangrove Cay, North Andros, New Providence) had fewer than ten buoys visit them over the course of our study period and were thus removed from our analyses. We had two metrics for dispersal potential between islands: minimum dispersal time and dispersal probability, which were significantly correlated with each other (ρ = −0.51, p < 0.001). The longest thalassochoric

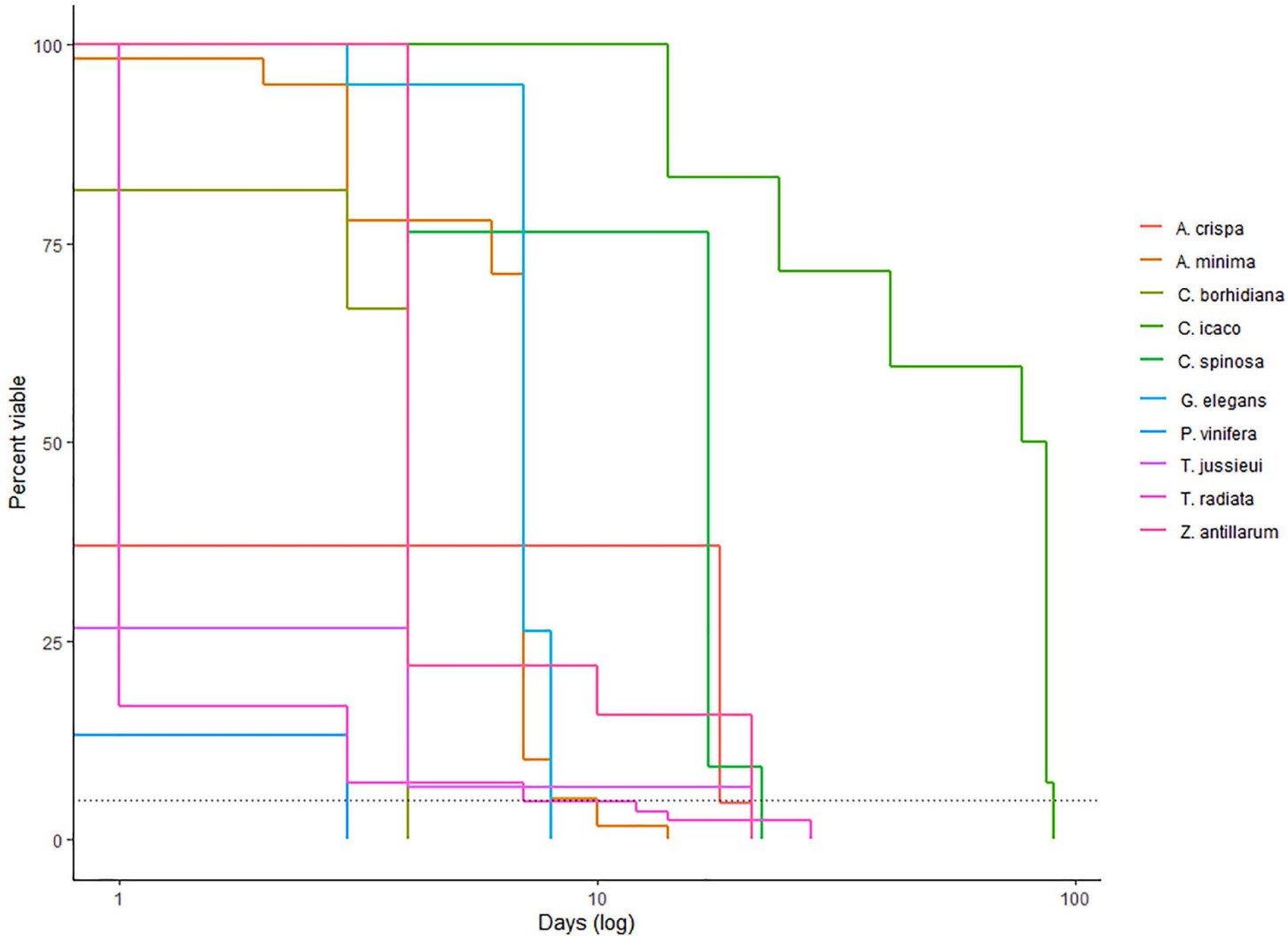

**Fig 2. Oceanic dispersal capacity of Caribbean fruits.** We found significant differences in oceanic dispersal capacity of our study species ($\chi^2 = 1215$, df = 13, p < 0.001). The best floating species was *Chrysobalanus icaco.* Four species (*Coccothrinax barbadensis, C. spissa, Gaussia attenuata, Syagrus amara*) all sank within a day and are thus not represented in this figure. The dotted line shows where 95% of fruits were no longer viable (i.e., had sunk or failed the Tetrazolium test), which we used as our 95th percentile viability metric.

transit time occurred from Barbuda to the Great Exuma Island (630 days), for which dispersal probability was ~4% (3 out of the 76 buoys departing Barbuda). By comparison, the thalassochoric transit time from Barbados to Isla de la Juventud, the most geographically distant pair of islands (2600 km), was less than the maximum viability time of *Chrysobalanus icaco* (90 days) at 88 days, with a dispersal probability of 7% (14 out of the 205 buoys departing Barbados). When comparing ocean-current–based dispersal metrics, both minimum dispersal time ($\rho = 0.89$, p < 0.001) and dispersal probability ($\rho = -0.35$, p < 0.001) were significantly correlated with geographic distance, but minimum dispersal time showed a stronger correlation (z = 18.31, p < 0.001).

In assessing the capacity of ocean-current dispersal potential for predicting species distributions, we found a significant, positive relationship between maximum and 95th percentile viability time and species range size for all study species (Fig 3: max: estimate (standard error) = 0.19 (0.086), z = 2.18, p = 0.03; 95th percentile: estimate (standard error) = 0.25

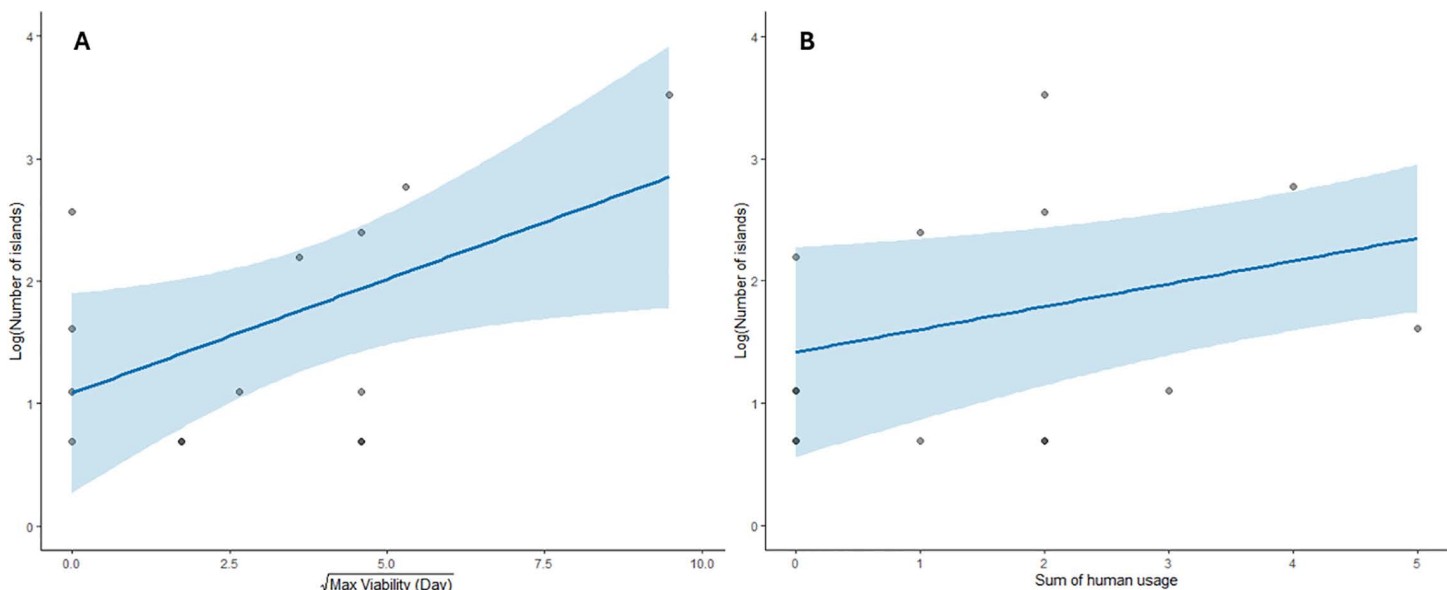

**Fig 3. Both oceanic dispersal capacity and the degree of human use were significant predictors of species range size (Maximum viability: z = 2.2, p = 0.03, Sum of human usage: z = 2.7, p = 0.006).** Range size (number of islands on which a species occurred) is shown on a log-scale, although the model was fit using the integer values and a Poisson error distribution. Maximum viability time was square root-transformed to increase normality..

(0.031), z = 8.08, p < 0.001). This was accompanied by a significant, positive relationship between species range size and the sum of human use categories (Fig 3: max: estimate (standard error) = 0.22 (0.082), z = 2.72, p = 0.006; 95th percentile: estimate (standard error) = 0.25 (0.075), z = 3.36, p < 0.001). Residual plots against predicted values for both sets of models showed no systemic patterns, and none of the models experienced convergence or singularity issues. When examining which islands each species occurred on, no species' distributions were significantly correlated with their level of ocean-current connectivity or geographic distance. In evaluating the relationship between community similarity and metrics of ocean-current dispersal (dispersal probability and dispersal time), we found no instances of significant correlation for either of the metrics. Additionally, we found no significant difference in correlation between floating and non-floating communities and metrics of ocean-current dispersal. We found similar results with geographic distance (S7 Table).

## Discussion

Long distance dispersal (LDD) is critical to connecting populations and shaping species distributions. While fleshy-fruited plants primarily depend on frugivory [56,57], these plants could also benefit from abiotic means of dispersal, such as via ocean currents [15]. Given precipitous declines in populations of biotic dispersal agents [6], such alternative means of LDD may be increasingly important for maintaining these connections. We demonstrated that some fleshy-fruited plant species can remain viable and floating in salt water for extended periods of time, allowing species such as *Chrysobalanus icaco* to disperse between the most geographically distant pair of islands in the Caribbean archipelagos. Additionally, we found that species that are better suited for dispersal by ocean currents tend to be found on more islands.

Historically, many fleshy-fruited plants are thought to have been dispersed across oceans by either being carried on floating vegetation [9] or after being consumed by large animals, which then transited across large bodies of water. In particular, the ability of giant tortoises to float across vast oceanic distances has been documented [58], and this ability is thought to have been responsible for pre-historic trans-oceanic dispersal of many plants [59,60]. As giant tortoises used

to populate the Caribbean [34], it is feasible that similar processes occurred in our study region. Due to defaunation, however, much biotic dispersal potential may have been lost, effectively stranding large fruits that are unable to float. An example from our study of a successful plant species despite these obstacles is *Chrysobalanus icaco,* which possesses fruits too large to be consumed by any extant birds in the Caribbean archipelagos [40]. However, it is also the species with the widest distribution in our study [31]. Given our results, we suggest that this phenomenon may be due in part to this species' capacity to float and stay viable for long periods of time.

This further expands on the role that ocean currents can play in the island biogeography of regions such as the Caribbean. Traditionally, biogeographic studies use geographic distance between islands as the metric of island isolation and connectivity [61]. As both our thalassochoric dispersal metrics were significantly correlated with geographic distance between islands, our results largely agree with the usage of geographic distance as the standard isolation metric in biogeographic studies. However, of the two ocean current metrics considered, the correlation with geographic distance was stronger for minimum dispersal time than for dispersal probability. This indicates that while fast dispersal between geographically close islands can occur, this may not always be representative of the total proportion of fruits that successfully disperse. Thus, ocean current dispersal can introduce a level of nuance in assessing levels of island connectivity, and this discrepancy could have significant biogeographical implications, as low immigration rates have been associated with the failure of plants to successfully establish on islands [57]. Such situations are also indicative of partial biogeographic barriers between areas of close geographic proximity. Within our study area, the Mona Passage between Puerto Rico and Hispaniola is well documented as a biogeographic barrier, with Baums et al. [62] finding genetically distinct populations of coral species on these two major islands despite their geographic proximity. Such results may be due to the swift, perpendicular current movement through the Mona Passage disrupting connectivity between this pair of islands.

Regarding fruit characteristics and ability to float, we curiously found that neither size nor density of fruits predicted floating potential. This conflicts with our expectations given our physical understanding of buoyancy through Archimedes' principle, wherein less dense objects are expected to float better [63]. This result indicates that plants such as *C. icaco,* which had the highest dispersal capacity, possess characteristics other than buoyancy that assist their thalassochoric dispersal ability. Heleno and Vargas [15] identified several characteristics that define the thalassochoric dispersal syndrome of plants, such as corky or fibrous seeds, the presence of air chambers, and other structures. Indeed, *C. icaco* possesses hollow seeds (S3 Fig), which were recorded to float even after its pulp was removed by extended exposure to salt water. However, as most fleshy fruits do not possess such characteristics, pulp traits may play more important roles. Given that density did not predict floating potential of fruits, we suggest that water permeability may be one trait that predicts how well fruits can float. Fleshy-fruited plants are known to possess different levels of water permeability [63], and if water can penetrate a fruit's exocarp and inundate the mesocarp, it could weigh down the fruit by displacing any air pockets and sink it, thus limiting its thalassochoric dispersal potential regardless of its seed traits. Therefore, we recommend future studies to examine both seed and pulp traits in assessing the adaptive traits of plants that may utilize ocean currents for dispersal.

Overall, our study represents a promising, initial examination of the associations between ocean current dispersal and fleshy-fruited plant distributions. Our results support previous studies' assertion of the importance of abiotic means of dispersal for plant species [23] and demonstrate that certain species are better suited to thalassochoric dispersal than others. Furthermore, we emphasize the biogeographic implications of ocean currents, which may supplement geographic distances in understanding how populations are connected in island regions [64]. It is important to note, however, that ocean current dispersal metrics were not effective in predicting distributions for any of our study species, and while we did find a significant relationship between species range size and oceanic dispersal capacity, we caution against making definitive conclusions given our small sample size. Furthermore, we did not find any significant difference in correlation between distributions of floating and non-floating species and ocean current dispersal metrics. As geographic distance was also not correlated with the distribution of our species, these results may be indicative of the difficulties in predicting

plant distributions in the region using any connectivity metric. It only takes a single successful LDD event to colonize a new island [5,6]. The rare and stochastic nature of such events may make it difficult to correlate their outcome (species distributions across islands) with mean connectivity metrics, whether they be purely distance based or measured empirically based on ocean currents.

Moreover, current plant distributions may be relics of historic dispersal and connectivity patterns. The Caribbean archipelagos possess diverse geological histories, with some islands potentially having been connected to either the mainland or to other islands due to changing sea levels and tectonic movements [65]. However, there is still debate regarding this topic, particularly that of the hypothesized land-bridge between the Greater Antilles and South America (GAARlandia) approximately 34 million years ago [66,67]. Regardless, while such geological histories could influence current plant distributions, the time scale of such geologic events suggest that over-water dispersal events could be more influential.

It is also possible that human activity in the region artificially influences plant connectivity and facilitates dispersal [68], thus obscuring efforts to understand the role of natural connectivity and dispersal in shaping plant distributions. This is supported by our results, where we found a significant and positive relationship between human usage and species distributions. For example, the study species with the largest distribution and greatest floatability (*C. icaco*) is used for both food and medicinal purposes [69,70]. However, it is not clear if human usage leads to larger species distributions or if humans are more likely to use plants that are already widely distributed. While we again caution against making any generalizations given our relatively small sample size, the relationship between plant range size and human usage presents an interesting topic of future study.

To convincingly link plant thalassochory with broader population dynamics, we recommend genetic studies, as have been conducted by Galindo et al. [20] for coral species. Such a study could use our metric of thalassochoric connectivity and compare it against empirical genetic data of different plant species, with the hypothesis that populations on islands with higher thalassochoric connectivity share greater genetic similarity. Another recommended approach is plant growth studies, where growth rates of seeds immersed in salt water for extended periods of time are directly compared against control seeds. While our study showed that seeds could be viable after extended periods of saltwater exposure, growth rate studies would help answer if salt water had significantly affected the seeds' overall performance and fitness. Previous research has shown that certain plants, particularly those that are not adapted for living on the coast, suffer from reduced growth and establishment rates with increased saltwater intrusion [71]. Therefore, growth rate analyses could further show which plants are more adapted for thalassochoric dispersal.

## Supporting information

**S1 Fig. Setup used to test for fruit floating potential in the lab.**
(PNG)

**S2 Fig. Flotation device used to assess the floating potential of fruits in the field.**
(PNG)

**S3 Fig. Results of tetrazolium test for *Chrysobalanus icaco*.** Viable seeds are stained red (left), while non-viable seeds are not (right).
(JPG)

**S4 Fig. Overview of buoy starting points in our region.** Yellow circles are starting points for all of our buoys.
(PNG)

**S5 Fig. Search areas (n = 217; yellow circles) around each of our study islands.**
(PNG)

**S6 Fig. Sample distribution (from Barbados to Dominica) of buoy transit times.** For all distributions, see Github link. (JPG)

**S1 Table. Particle dispersal probability between every island pair.**
(CSV)

**S2 Table. Minimum dispersal time between every island pair.**
(CSV)

**S3 Table. Sample species connectivity matrix (for *Chrysobalanus icaco*) for every island pair.** See Github link for all connectivity matrices.
(CSV)

**S4 Table. Sample species distribution matrix (for *Chrysobalanus icaco*) for every island pair.** 1 = joint presence. NA = joint absence. 0 = differing absence/presence. See Github link for all distribution matrices.
(CSV)

**S5 Table. Community similarity matrix using the Sørensen similarity index for species that can float for more than the median 95$^{th}$ percentile viability time (7 days).**
(CSV)

**S6 Table. Community similarity matrix using the Sørensen similarity index for species that can float for less than the median 95$^{th}$ percentile viability time (7 days).**
(CSV)

**S7 Table. Statistics for assessing and comparing correlation between floating and non-floating communities, ocean current dispersal metrics, and geographic distance.**
(XLSX)

## Acknowledgments

We would like to thank Montgomery Botanical Center and Fairchild Botanical Garden for permission to use their plant collections. Additionally, Montgomery Botanical Center allowed us access to their grounds to perform our floating field trials.

## Author contributions

**Conceptualization:** Seokmin Kim, Sofany Montoya, Christina Chavez, Joanna Tucker Lima, Donald Olson, Christopher Searcy.

**Data curation:** Seokmin Kim, Fabio L. Tarazona-Tubens, Christina Chavez, Joanna Tucker Lima, Christopher Searcy.

**Formal analysis:** Seokmin Kim, Donald Olson, Christopher Searcy.

**Investigation:** Seokmin Kim, Sofany Montoya, Fabio L. Tarazona-Tubens, Christina Chavez, Joanna Tucker Lima, Donald Olson, Christopher Searcy.

**Methodology:** Seokmin Kim, Sofany Montoya, Fabio L. Tarazona-Tubens, Christina Chavez, Joanna Tucker Lima, Donald Olson, Christopher Searcy.

**Project administration:** Seokmin Kim, Joanna Tucker Lima, Christopher Searcy.

**Resources:** Seokmin Kim, Fabio L. Tarazona-Tubens, Joanna Tucker Lima.

**Software:** Seokmin Kim.

**Supervision:** Donald Olson, Christopher Searcy.

**Validation:** Seokmin Kim, Donald Olson, Christopher Searcy.

**Visualization:** Seokmin Kim, Christopher Searcy.

**Writing – original draft:** Seokmin Kim, Donald Olson, Christopher Searcy.

**Writing – review & editing:** Seokmin Kim, Sofany Montoya, Fabio L. Tarazona-Tubens, Joanna Tucker Lima, Donald Olson, Christopher Searcy.

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
