## [Decision Letter · Decision Letter 0]

29 Apr 2025

PONE-D-25-10991Ocean current dispersal could benefit fruited plants in the CaribbeanPLOS ONE

Dear Dr. Kim,

Thank you for submitting your manuscript to PLOS ONE. After careful consideration, we feel that it has merit but does not fully meet PLOS ONE’s publication criteria as it currently stands. Therefore, we invite you to submit a revised version of the manuscript that addresses the points raised during the review process.

Your manuscript has been reviewed by three expert reviewers who have provided  extensive feedback to improve its quality. Firstly, the introduction should be restructured to change how the study is framed. Some ideas and concepts need to be clarify. For example: 1) differencies  between the contribution of  thalassocory to range expansion and the maintenance of the gene flow among populations; 2) use the term ”fleshy-fruited” rather than “fruiting”, etc. On the other hand, the idea that thalassochory may replace animal dispersers due to defaunation seems controversial.

There are also comments about the variables used to analyse the relationship between the species range and the floatability. Statistics also need a revision to improve its clarity. Furthermore, reviewers suggest some changes in the statistical analyses that have to be considered and they also drawn your attention on the high number of missing data in your data sets. Please, consider how the lack of independence in your data due to the phylogenetic relationship among the plant species might impact on your results.

The reviewers and I consider that all these problems can be solved with a thorough revision. So, I encourage you to address all these comments and resubmit your manuscript.

We look forward to receiving your revised manuscript.

Kind regards,

Vicente Martínez López

Academic Editor

PLOS ONE

Additional Editor Comments (if provided):

Reviewers' comments:

Reviewer's Responses to Questions

**Comments to the Author**

1. Is the manuscript technically sound, and do the data support the conclusions?

Reviewer #1: Partly

Reviewer #2: Yes

Reviewer #3: Partly

2. Has the statistical analysis been performed appropriately and rigorously? 

Reviewer #1: Yes

Reviewer #2: Yes

Reviewer #3: No

3. Have the authors made all data underlying the findings in their manuscript fully available?

Reviewer #1: No

Reviewer #2: Yes

Reviewer #3: No

4. Is the manuscript presented in an intelligible fashion and written in standard English?

Reviewer #1: Yes

Reviewer #2: Yes

Reviewer #3: Yes

5. Review Comments to the Author

Reviewer #1: I enjoyed reading the manuscript by Kim et al. exploring the thalassochoric potential of Caribbean fleshy fruits and their correlates with distribution range, island distance and estimates of oceanic connectivity. The paper explored a clearly understudied issue and contains a lot of high-quality data (I particularly liked the ocean connectivity analysis based on tagged buoys from NOAA) and solid analysis.

I have several important concerns about the framework in which some results are presented and discussed but I hope most can be solved with an in-depth revision. I also list a large number of minor corrections and suggestions that I hope can further assist the authors in improving their valuable work.

General points

I feel that the paper mixes a bit the role of thalassochory for colonisation of new islands (natural range expansions) and the role of dispersal for maintain gene flow and viable populations. For example, thalassochory is argued to be particularly important in the context of current defaunation. This rational isn’t completely wrong, but I think that there should be a better separation between the conservation and the biogeography issues, as the latter occurs on a much slower time scale. For reference, natural colonization’s of new species arriving to an island (those that can increase range) is a process that occurs on average only once every several thousand years (so this issue is largely decoupled from the issue of defaunation/conservation).

I’m puzzled (probably like the authors) that there is no effect between fruit density and flotation capacity. Doesn’t that go against the laws of physics? I guess that objects should float on liquids only if they have a lower density than the liquid. So, density is by definition related to flotation potential. There is some suggestion that water absorption could disrupt flotation (L375), but I’m not sure whether this can be the case as water cannot make any object denser than water itself. Maybe I’m not getting my physics right (sorry if that’s the case), but I think this issue deserves a more solid explanation. Actually, it would also be positive to explain beforehand why this is an important research question (why isn’t just a straightforward physical definition that all fruits denser than water will sink). Maybe impulsion, salinity and superficial tension forces need to be invoked here.

The median 95th percentile viability time was used to classify species as “high” (>7 days) or low (<7 days) thalassochorous potential, but it wasn’t completely clear to me what this number actually means. Is this the median of all values within the 95th percentile (i.e. an estimate of a central measure of flotation) or is it the median of the values that represents the max. 95th percentile (i.e. an estimate of an extreme/near-maximum flotation capacity?). I think it makes more sense if it is a central measure, but please just clarify/confirm.

The relationship between species range (#islands) and viability while floating is only very very marginally significant at α=0.1 (p = 0.096) and most readers will probably consider this a non-significant result. This “extreme marginality” should be more clearly stated in the abstract and discussion. However, I wander (hope) if the relationship will be better with other measures of viability estimation (see previous comment/doubt about the median). Also, please add the result of the test directly into Fig. 3 to make life easier for the reader.

The abstract doesn’t say anything about viability and says very little about the current’s connectivity estimations (which are two important parts of the manuscript). I suggest the authors find space in the abstract to mention briefly these two points.

The fact that some islands have been connected by land bridges is only passively presented in the discussion, but it forms a strong constraint (and explanation) to the lack of an association between thalassochory connectivity and island distance. I feel this caveat should be assumed more clearly much earlier and probably it jeopardizes some of the analyses as it really doesn’t make much sense to assume the relevance of thalassochory for explaining the presence of a plant in two islands that have been connected for thousands of years in the past.

Maybe I missed something, but I couldn’t find the supplementary tables mentioned in the Supplemental materials word document.

Specific issues:

L35 its probably easier and shorter to say simply: 2600 km, rather than 2.6×10^3 km

L36, L98 “floating better” sounds a bit weird to me has it could be understand has having a lower fraction of the seed inside the water. Probably better replacing “better” by something like: longer or greater floating capacity.

L52 [3. 4]. -> [3, 4].

L56 “plays a potentially interesting role” not great choice of words. Maybe better “important” or “relevant”

L65 I’m not familiar with the term “turbulent dispersal”, maybe clarify.

L67 The study of Cryptocarpus pyriformis is also in the Caribbean? If not, consider mentioning the Archipelago

L77 has evolved traits facilitating this process -> has evolved traits to facilitate this process

L79 more large -> larger

L113 Figure 1 should be cited here

L117 [32 33]) -> [32, 33]).

L125 consider: utilize -> benefit from

Table 1 add a “.” After “diam” (diam.) and “max” (max.)

L138 It sounds more logical to cite the Supplementary figures as: (Fig. S1) instead of (S1 Fig), but this might be a journal style, so I wont comment on that further.

L145 “free-moving/flowing salt water”. Not very clear, was this a bay? Or a swimming pool filled with saltwater and a circulation pump? Please clarify.

L156 So this is a paired t-test for the differences for the 5 species used on both methods? Or a t-test for different species used in each method? Please add a bit more detail so that the reader can understand what was done. If there is no space in the main manuscript, this could be a small Supplement.

L159 [39-40] -> [39, 40]

L199 The word “arbitrarily” seems a bit excessive here. Consider instead something like: “…44 major Caribbean islands (here considered as those larger than 100 km^2) “.

L236 (Fig 2) -> (Fig. 2) (find and replace throughout manuscript Fig -> Fig.)

L176, L243 Uniformize the way packages are written (with/without quotation marks; with/without italics)

L270, L283 Consider: there is significant variation in thalassochoric… -> …there are significant differences in thalassochoric

L271 You can abbreviate genus names after first mention.

L279 Consider: with 33 islands -> present in 33 out of 44 islands …

L279 indicate table with these results

L300-305 This section could be written with a lot less words (avoiding some repetitions).

L311 “(insert stats).” Good idea :)

L311-319 This section is hard to read and carried a few weird and not very fluent text. Please try to improve readability.

L332 Best to highlight here that this relationship is very poorly supported (P=0.096)

L410 Could this also be tested by contrasting the presence of plants in inhabited and uninhabited islands?

Reviewer #2: The manuscript entitled "Ocean current dispersal could benefit fruited plants in the Caribbean" by Seokmin Kim et al. evaluates the ability of 14 plant species native to the Caribbean islands to float and survive on salt water as a baseline to assess their potential for dispersal by ocean currents. The ability of ocean currents to shape and predict the distribution of these species is also assessed. The results indicate that these species, which are not a priori specialised for sea dispersal, differ in their ability to float and survive, and that this ability is directly related to the number of islands on which these species occur. However, no relationship was found between ocean currents dispersal metrics and species distribution. Assessing population connectivity and species dispersal ability is important to understand the degree of species isolation, particularly in restricted and isolated territories such as islands, in order to take effective conservation action and prioritize efforts and resources.

This study nicely combines an experimental approach and analytical techniques to assess the potential of species to be dispersed by ocean currents. The authors assess the floatability and viability of fruits in natural conditions (using an ingenious device), which is a valuable information to accurately evaluate whether a species is able to float and survive the minimum time necessary to be transported from one island to another. For this reason, I believe this manuscript has great potential to be published. However, I have two main concerns. My first concern has to do with the way this study has been framed. The interesting point about assessing thalassochory in these 14 species is that these species have fruit specializations that attract animals as dispersal vectors, and for which the sea was not thought to be the main dispersal agent. This decoupling between fruit specialization and long-distance dispersal has been extensively discussed in the literature (e.g. Heleno & Vargas, 2015; Higgins et al., 2003; Nathan, 2006), and the manuscript will be enriched if it is framed in this context. Following this suggestion will mean restructuring the introduction (see my suggestion below) and upgrading the discussion by including a broader topic of global interest. In addition, the bibliography cited in the discussion is rather poor. Regarding the role of ocean currents as dispersal vector, the authors can greatly improve the discussion of their results in light of previous studies that show their importance in transporting non-specialized organisms (e.g. lizards, Calsbeek & Smith, 2003), as well as specialized organisms (e.g. mangroves, Hodel et al., 2018; Ipomoea pes-caprae, Miryeganeh et al., 2014), studies conducted in the Caribbean region and in other parts of the world. My second concern relates to the statistical analyses. The text is somewhat difficult to follow, and the use of some terms adds confusion (see my comments below). The authors rely mainly on Mantel tests, whose statistical power has been questioned. I believe that other models would allow the authors to test different hypotheses simultaneously (e.g. the role of sea dispersal, geography and human use/impact) and to draw stronger conclusions (see my suggestions below).

Regarding the main text, the first thing I find quite confusing is the term "fruited plants", as it broadly means plants with fruits. After reading the manuscript I believe the authors are referring to plants that produce fruits with endozoochorous characteristics in the sense Heleno & Vargas (2015), i.e. edible fruits with fleshy pulp and/or bright colours that attract animals that may eat them. This is the first thing the authors should make clear, as this study is clearly focused on species with characteristics that make them good candidates to be mainly dispersed by animals. Then, I would recommend that the introduction focus first on the dispersal of plants with endozoochorous fruits (hereafter fleshy fruited plants for simplicity), and the problems of defaunation specially in the long-distance dispersal (LDD) of this species. Then I would focus on islands and LDD, and the decoupling between the dispersal syndrome and the dispersal vector (see Heleno & Vargas, 2015). Finally, I would introduced the potential of fleshy fruits to be dispersed by ocean currents (Esteves et al., 2015) that would ultimately lead to the main knowledge gap that this paper addresses and to the main questions. If I have misunderstood the main species selection criterion for this study (i.e. plants producing endozoochorous fruits), the authors should clearly state their criteria for plant selection. As this determines the whole text and its main focus.

I find the methods section a bit confusing because of all the terminology, the complexity of the methods and the many tests used. In particular, from line 235 to line 266, I strongly recommend that the authors make an effort to rewrite it for the sake of clarity. In this paragraph the same or similar terms are used for different things. First, the authors talk about thalassochoric connectivity (L. 235), which is species specific. Then, it is tested the correlation between geographic distance and thalassochoric dispersal potential (L. 242), but this latest concept refers to the two thalassochoric dispersal metrics (Tables S1 & S2), which have nothing to do with the species but with the ocean currents themselves. It continues with the evaluation of the predictive power of the thalassochoric dispersal potential on species distributions (three tests; L. 249), but the two first tests are species specific and they use either the viability metrics or the species connectivity matrices (not the dispersal metrics Tables S1 & S2), and the third test refers to the ocean current connectivity matrices as the dispersal probability and minimum dispersal time (dispersal metrics Tables S1 & S2), not the species thalassochoric connectivity.

My recommendations for improving the analyses are:

(1) GLM fitted using a Poisson family with the number of islands that each species occur on (species range size) as response variable, and the species viability metrics (maximum / 95th percentile) and human use as predictor variables [no.islands ~ viability + human use]. Instead of the linear regression of L. 250-251 and L. 263-265.

(2) Generalized dissimilarity models (GDM) to investigate the role of geography, ocean currents and island attributes (e.g. human population, island size, island age, climatic values per island) in influencing floristic similarity between islands (Sørensen index) (Ferrier et al., 2007; Mokany et al., 2022). These models allow the use of a floristic similarity matrix as a response variable and a combination of distance matrices and raw data as predictors. In this way, the authors could also assess the influence of island characteristics, such as island size or age, which according island biogeography theory are of great importance in shaping island biodiversity, but also human-related island characteristics such as population density or some island metric of human impact (L. 400-402). At this point, one thing the authors need to be careful about is the high correlation between thalassochoric dispersal metrics and geographic distance. This may affect the GDM results and perhaps it would be worthwhile to additionally fit GDMs with geography (+ island characteristics) and dispersal metric (+ island characteristics) separately.

Regarding the data, particularly Supp. Tables 1, 2, 5 and 6, how have NAs been handled in the analyses? I understand that the connectivity measures with NA values are those with no buoys connecting two islands, which makes me wonder if they have been converted to zero for the probability of dispersal and to a very high number for the minimum dispersal time. In this way, these "unconnected" island pairs are included in the analyses. However, the procedure followed by the authors is not clear in the text, apart from the mention of removing seven islands from the analysis (L. 292-293). Moreover, I do not understand why there are NAs in Supp. Tables 5 and 6 (Sørensen index). If two islands have no species in common, should not it be zero? These concerns should be addressed as they directly affect the results and statistical power of the analyses.

Other comments:

Title: I would recommend the authors to think of a more catchy title, as the one they have proposed suggests that no conclusions are drawn about the role of ocean currents as a dispersal vector, which this is not very appealing.

L. 35: Replace 2.6 x 10³ km by 2600 km, as it is not a super big number and it will improve readability.

L. 56-57: Ocean currents is a vector, thalassochory is the dispersal mechanism, please rephrase. In addition, there are not many vectors capable to transport propagules to islands, they are either wind, ocean currents or flying animals, so the authors can confidently reformulate the sentence into a statement: instead of "ocean currents are thought", "ocean currents are one of the main LDD vectors".

L. 65: I do not understand what this has to do with the role of ocean currents in connecting populations, I would remove the last part of the sentence "actual connectivity is limited by turbulent dispersal" unless the authors find a better way to integrate it into the flow of the text.

L. 70: I would not say that the ability for fruits to float and disperse is widely documented. I would rather say that it has been documented for some species with special emphasis on coastal species.

L. 71-73: Is this referring to long-distance dispersal to islands? If so, there are only three other ways for plants to overpass the sea barrier. Apart from what the authors specify here, propagules can arrive on islands attached to the fur/feathers/skin of flying vertebrates (epizoochory) (e.g. Aoyama et al., 2012). The authors should clarify here if they are talking about LDD to islands and if so, specify all the other ways of LDD for plants to reach islands (anemochory, endo- and epizoochory). And then remove "thalassochory is usually not the only method of dispersal by plants" as it is obvious. I recommend to rephrase the whole paragraph.

L. 116: "main" instead of "mean"?

L. 127: why relying on existing dataset if the authors worked directly with the fruits? Why not measuring them before the experiments?

Table 1: it would be interesting to include the maximum number of days the fruits were floating per species. I am wondering if this value is decoupled with seed viability in some species, for example, due to aborted seeds, these "empty" seeds will favour higher fruit floatability.

L. 190: "[" 33, 46, 47].

Figure 2 should indicate the species to which each curve belongs. In my opinion, the colour would not be necessary if an horizontal line indicating the 95% of floating viable fruits is included. The y-axis should be labelled "Percentage of viable floating fruits". If the authors have chosen to present the x-axis in logarithmic scale only for better visualization of all lines, this should be mentioned in the figure caption. I have a suggestion regarding this x-axis transformation, it is true that it would be difficult to differentiate all the early sinking species lines in a natural scale, but it is also easier to understand. So my suggestion is to do a first plot with the x-axis in natural scale and then do either a zoom in and show only a subset of days (e.g. 10 or 20 days) or show a second plot with the x-axis in logarithmic scale.

L. 250: "we conducted a linear regression", if the response variable was the number of islands (which should be indicated), this variable follows a Poisson distribution, and hence a GLM should be performed instead.

L. 311: "(insert stats)"

L. 318-319: already mentioned in L. 309-311.

Figure 3: x-axis label should indicate that it represents the maximum viability time. Why the square root? If this value was transform for fitting the model, it should be specified in the main text. Y-axis label should be "Log (number of islands)", and specify in the figure caption that number of islands is the "range size". Was the logarithmic transformation only for representation purposes only?

L. 332: I would avoid the term 'adapted' as it implies an evolutionary relationship between a particular biotic/abiotic factor and a species trait. In this case, we do not know whether the higher floatability and viability of certain species has evolved in response to a more efficient sea dispersal, or whether it is an unintended consequence of the particular evolutionary history of the species. I would recommend using "better suited for dispersal by ocean currents" instead.

L. 378: "hollow seeds" can be considered as seed with an air chamber, in line with Heleno & Vargas (2015) (L. 369-371), then it would contradict the statement made in L. 371-372.

Supp. Table 7: I could not find it in Github (https://github.com/sxk1332/floating_fruits/tree/main/Supp_materials)

References

Aoyama, Y., Kawakami, K., & Chiba, S. (2012). Seabirds as adhesive seed dispersers of alien and native plants in the oceanic Ogasawara Islands, Japan. Biodiversity and Conservation, 21(11), 2787–2801. https://doi.org/10.1007/s10531-012-0336-9

Calsbeek, R., & Smith, T. B. (2003). Ocean currents mediate evolution in island lizards. Nature, 426(6966), 552–555. https://doi.org/10.1038/nature02143

Esteves, C. F., Costa, J. M., Vargas, P., Freitas, H., & Heleno, R. H. (2015). On the Limited Potential of Azorean Fleshy Fruits for Oceanic Dispersal. PLOS ONE, 10(10), e0138882. https://doi.org/10.1371/journal.pone.0138882

Ferrier, S., Manion, G., Elith, J., & Richardson, K. (2007). Using generalized dissimilarity modelling to analyse and predict patterns of beta diversity in regional biodiversity assessment. Diversity and Distributions, 13(3), 252–264. https://doi.org/10.1111/j.1472-4642.2007.00341.x

Heleno, R., & Vargas, P. (2015). How do islands become green? Global Ecology and Biogeography, 24(5), 518–526. https://doi.org/10.1111/geb.12273

Higgins, S. I., Nathan, R., & Cain, M. L. (2003). Are long-distance dispersal events in plants usually caused by nonstandard means of dispersal? Ecology, 84(8), 1945–1956. https://doi.org/10.1890/01-0616

Hodel, R. G. J., Knowles, L. L., McDaniel, S. F., Payton, A. C., Dunaway, J. F., Soltis, P. S., & Soltis, D. E. (2018). Terrestrial species adapted to sea dispersal: Differences in propagule dispersal of two Caribbean mangroves. Molecular Ecology, 27(22), 4612–4626. https://doi.org/10.1111/mec.14894

Miryeganeh, M., Takayama, K., Tateishi, Y., & Kajita, T. (2014). Long-Distance Dispersal by Sea-Drifted Seeds Has Maintained the Global Distribution of Ipomoea pes-caprae subsp. Brasiliensis (Convolvulaceae). PLOS ONE, 9(4), e91836. https://doi.org/10.1371/journal.pone.0091836

Mokany, K., Ware, C., Woolley, S. N. C., Ferrier, S., & Fitzpatrick, M. C. (2022). A working guide to harnessing generalized dissimilarity modelling for biodiversity analysis and conservation assessment. Global Ecology and Biogeography, 31(4), 802–821. https://doi.org/10.1111/geb.13459

Nathan, R. (2006). Long-Distance Dispersal of Plants. Science, 313(5788), 786–788. https://doi.org/10.1126/science.1124975

Reviewer #3: This study examines how long (if!) the fruits of some fleshy-fruited species in the Caribbean stay afloat in saltwater, and what this means for seed viability. This is an important question for long-term vegetation dynamics in any island-rich region – and thus especially so in the Caribbean, with its myriad islands and archipelagoes.

I really liked that the authors used two complementary methods to investigate floating potential (even if not possible for all species); this is done much too rarely in such studies.

While topically interesting and with valuable data, the study in its current form suffers from several major issues:

1) The most important issue is the phylogenetic non-independence in your data: A whopping ten out of the 14 study species are from the same family (Arecaceae; palms), and three of these from the same genus. Given how similar the fruits of many palm species are, this strong non-independence of your study species needs to be discussed (in terms of how it could/could not impact your results & their interpretation).

This is especially important for your analyses, where you should really include family (or at least genus) as a random effect (or at least fully explain why this is not necessary).

2) I don’t buy the argument that thalassochory is becoming increasingly important in light of the terrestrial defaunation of large-bodied dispersers. The former is most concerned with inter-island dispersal, while the latter is almost entirely about intra-island dispersal.

For the day-to-day ecology of any slightly large island (where ‘inland’ > ‘coastal’ ecosystems), what matters most is the loss of (formerly common) intra-island dispersal, rather than the maintenance of (comparatively uncommon) oceanic transport of seeds from fleshy-fruited plants. I would suggest toning down the quite strongly worded conservation implications for island ecosystems in general, and stick to what it means for coastal plant communities.

3) There are some issues with the origin and use of ‘traits’ in the MS that need to be addressed. The authors give [35] as reference for the size and weight (“We relied on an existing dataset [35] for information regarding the size and mass of each fruit.”) – but this reference is an in-prep study by an author not on the current MS, and should not be used/cited in this way.

Please cite these measurements as unpublished data, and make sure to acknowledge this person for the use of it. (and see my point 5, below, for how this affects Table 1, too).

4) Throughout the MS the writing is quite ‘loose’/unclear when it comes to the use of ‘plant’, ‘species’, and ‘fruit’. I have tried to catch the worst of them & suggest rewordings (in the section on minor issues, below), but please make sure that you are as specific as possible, to make it easier for the reader to follow your arguments. In particular, use ‘fleshy-fruited’ instead of ‘fruiting’.

5) Table 1 has several issues that need to be addressed:

- Please add a column with family of the species; to many readers this will be much more informative to get a quick idea of the fruit type (e.g., that ten of the 14 study species are palms!) than only giving the speices name.

- It is unconventional to rank study species in the methods by results given in the results section; It should really be done by plant family (& alphabetically within family) as is usual for such studies.

- For anyone looking at Table 1, they will currently almost certainly make the assumption that these measurements come from the fruits you used in your study (column 5, ‘initial count’). Please make sure to include the info for the origin on these trait values in the legend, too.

- It makes no ecological sense to give the mean values in columns 2-3 with an accuracy of two decimal points, please simplify to one.

- The title is a bit misleading, since only 3 out of 7 columns are actualy traits; I suggest renaming this to simply say ‘Study species’.

6) Figure 2 has some issues that need to be addressed:

- The legend: While you use survival curves to show your results, ‘survival’ is not what you measure/show here, but rather floating capability. Also, the first line reads ‘of all species used in our study’, but the last line says that the seeds of four that sank within a day are not given.

Please change the first line to something more meaningful/simple, e.g., “Fig 2. Floating capacity of fruits”.

- Giving each curve a colour based on the floating capacity is odd, as this information can be read/seen directly in the figure, anyway. This also currently hides the biological meaningful information about which survival curve belongs to which species.

Instead, please give each species’ curve a different colour and label the individual survival curves – e.g., with a number somewhere along its length – and list the corresponding species in the legend (or to the right of/below the main figure).

7) Throughout the MS, I suggest simplifying ‘fruit traits’ to simply be referred to as ‘size and weight’ (as these are the only ones you recorded; for many dispersal ecologists, ‘fruit traits’ usually refers to more than just these two.

8) L311 – ‘(insert stats’) – yes, please!

Minor issues and corrections:

L27 & again in L55 – suggest using a simpler wording, “Thalassochory, or dispersal by ocean currents, …”

L29-32 – a very long and winding sentence, suggest splitting up into two.

L30 – correct ‘fruiting’ to ‘fleshy-fruited’.

L31 – insert comma, “…potential, and…”

L33 – you have just defined which plants you study in the previous sentence; hence suggest simplifying here by deleting, “…of plants in the region”.

L34 – suggest using, “…feasibly being able to float…”

L35 – there is no need to use exponential notation here, just say 2,600 km…

L36 – reword to, “...for fruits with longer floating times…”

L43 – correct to ‘archipelagos’, to fit with usage in rest of MS.

L44 – insert comma, “…saltwater, and 2)…”

L45 – correct ‘fruiting’ to ‘fleshy-fruited’.

L52 – reword to, “…and have implications for both conservation management and biogeography”.

L52 – correct to, “Islands are classic study systems…”

L69 – This sentence is very unclear/uses wrong terms – suggest rewriting to, “Succesful thalassochory requires the fruit or seed to float and the seed to stay viable while at sea – both of which have been widely documented for plants in coastal regions and islands”, or similar.

L75 – there is only species in the genus Cocos; correct to ‘Cocos nucifera’)

L76 – insert comma between ‘dispersal and’.

L79 – correct to ‘large-fruited species’

L83 – correct ‘fruiting’ to ‘fleshy-fruited’.

L84 – insert comma after ‘dispersal [27]’

L90 – insert comma after ‘…than others’

L93 – insert comma after ‘archipelagos’

L94 – simplify to, “We address the following…”

L94 – reword to, “Do the fruits of some plant species…”

L96 – correct ‘fruiting’ to ‘fleshy-fruited’.

L98 – reword to, “…plants with fruits that float better…”

L110 – correct to ‘eddy-dominated’

L112 – insert comma after ‘gyre’.

L129/280/282 – please use ‘records/recorded’ instead of ‘observations/observed’ (also rename corresponding column in Table 1 to reflect this)

L173 – correct ‘fruiting’ to ‘fleshy-fruited’.

L250 – move contents in parenthesis to appear after ‘…viability metrics’.

L251 – delete ‘size’; species range is already a spatial measure on its own.

L257 – insert comma after ‘…days)’

L262 – insert comma after ‘genus level’

L263 – correct to “each species’ range size”

L263 – delete the first ‘by’ and insert commas before and after ‘first’, to read, “We did so, first, by…”

L264 – insert commas before and after ‘second’.

L276 – correct ‘fruiting’ to ‘fleshy-fruited’.

L304 – correct to ‘more strongly correlated’

L327 – substitute ‘via’ for ‘through’.

L329/L333/L374/L384 – correct ‘fruiting’ to ‘fleshy-fruited’.

6. PLOS authors have the option to publish the peer review history of their article (what does this mean?). If published, this will include your full peer review and any attached files.

Reviewer #1: No

Reviewer #2: **Yes:** Yurena Arjona

Reviewer #3: No

---

## [Author Response · Author response to Decision Letter 1]

23 Oct 2025

Please see attached response for details.

---

## [Decision Letter · Decision Letter 1]

17 Dec 2025

PONE-D-25-10991R1Testing the potential of Caribbean fleshy-fruited plants to disperse using ocean currentsPLOS One

Dear Dr. Kim,

Thank you for submitting your manuscript to PLOS ONE. After careful consideration, we feel that it has merit but does not fully meet PLOS ONE’s publication criteria as it currently stands. Therefore, we invite you to submit a revised version of the manuscript that addresses the points raised during the review process.

The reviewers and I acknowledge the excellent work carried out by the authors in addressing all the comments raised during the first round of revision. The manuscript has improved significantly. However, the reviewers have identified some remaining issues that should be addressed before the manuscript can be accepted. I would be pleased to review a revised version of the manuscript once the authors have considered and responded to the reviewers’ concerns.

We look forward to receiving your revised manuscript.

Kind regards,

Vicente Martínez López

Academic Editor

PLOS One

Journal Requirements:

Reviewers' comments:

Reviewer's Responses to Questions

**Comments to the Author**

1. If the authors have adequately addressed your comments raised in a previous round of review and you feel that this manuscript is now acceptable for publication, you may indicate that here to bypass the “Comments to the Author” section, enter your conflict of interest statement in the “Confidential to Editor” section, and submit your "Accept" recommendation.

Reviewer #1: All comments have been addressed

Reviewer #2: (No Response)

Reviewer #3: All comments have been addressed

2. Is the manuscript technically sound, and do the data support the conclusions?

Reviewer #1: Yes

Reviewer #2: Yes

Reviewer #3: (No Response)

3. Has the statistical analysis been performed appropriately and rigorously? 

Reviewer #1: Yes

Reviewer #2: No

Reviewer #3: (No Response)

4. Have the authors made all data underlying the findings in their manuscript fully available?

Reviewer #1: Yes

Reviewer #2: Yes

Reviewer #3: Yes

5. Is the manuscript presented in an intelligible fashion and written in standard English?

Reviewer #1: Yes

Reviewer #2: Yes

Reviewer #3: Yes

6. Review Comments to the Author

Reviewer #1: I feel the authors satisfactorily addressed the concerns raised by the reviews which contributed to a clearer and stronger manuscript. I therefore believe the manuscript is valid contribution to the field and I would like to congratulate the authors for their nice study. I have only a few specific points for consideration that I outline below.

Specific comments

L85-86, I’m not sure if the 3 references cited in this sentence actually “demonstrate” that this mechanisms work for inter-island seed dispersal as neither of them managed to record the origin, vector and arrival of o colonizing propagule. We know that these 3 mechanisms are the potential ones because there are the only ones that can promote these movements, because they have been demonstrated to work on another contexts (not between islands), and because there is indirect probabilistic evidence suggesting that they are likely effective vectors. Please revise the term “demonstrated”, and also check that the references are the correct one (e.g. the paper dealing with the dispersal of the coconut palm is indicated as a demonstration of epizoochory). Also correct “..”

L96 The concept of “artificial introductions” is unclear. Consider replacing with anthropogenic introductions or a similar term.

L284 correct “..”

L309 consider: visit -> visiting

L435443 I’m not fully convinced that “growth rate” is a particularly informative variable to include in future studies. Even if plants exposed to salty environments grow more slowly due to physiologic constraints, they can still promote gene flow and new colonization events (only slower). Moreover, as the current manuscript is not about growth rates, this seems like a bad choice of topic to close the manuscript.

Reviewer #2: The revised version of the manuscript, now entitled "Testing the potential of Caribbean fleshy-fruited plants to disperse using ocean currents" by Kim et al., has improved compared to the first version I read. I appreciate the effort made by the authors in addressing all of the reviewer's concerns. However, I believe further work is still needed on building the background to provide a clearer framework for the study, and on presenting the statistical results.

More care should be taken in relating the topics of the effect of defaunation on fleshy fruit dispersal and the long-distance dispersal (LDD) of plants. Long-distance dispersal is a rare process in nature compared to dispersal over shorter distances (long tail of the species' dispersal kernel). Different dispersal vectors may play a role in this process, and these vectors do not necessarily align with plant dispersal specializations (non-standard means of dispersal, see e.g. Nathan 2006; Nogales et al. 2012; Vargas et al. 2012). This decoupling between the plant dispersal syndrome (in this case, fleshy fruits, that are indicative of an endozoochorous syndrome) and the dispersal vector opens the door to investigating how well suited are plant diaspores to be dispersed by non-standard vectors (e.g. fleshy fruits dispersed by sea). However, throughout the text, the authors sometimes give the impression that they are referring to dispersal as a deterministic process (e.g. L. 341-342), and that plants only look for another appropriate vector when the standard vector is missing (e.g. sea dispersal as an alternative to endozoochory in the absence of frugivores). I find the defaunation phenomenon an interesting motivation to assess the ability of fleshy fruits to be dispersed by other means, like sea dispersal. However, stronger evidence of the effect of the decline of frugivores on LDD, especially on islands, is needed. As another reviewer pointed out, defaunation may pose a significant challenge to inland plant dispersal (over short-medium distances) but not to crossing the sea barrier between islands (LDD). In line with my suggestion about LDD and plant dispersal syndromes, the authors can use the introduction of Esteves et al (2015), where the authors assessed the potential of Azorean fleshy fruits to be dispersed by sea currents, as a reference.

Another concern I have, especially after reading the Discussion, is the use and interpretation of 'population connectivity' in the framework of this study. This study is looking at the species presence/absence in the Caribbean islands, which must indeed be the result of some connection between islands. However, this does not mean that the connection is permanent or detectable nowadays. For example, an island could be colonized by a plant species after a single successful event of LDD followed by a complete isolation. Alternatively, a different regime of sea currents in the past might have connected two currently isolated islands. Presence/absence data cannot rule out the role of sea dispersal (or any other kind of dispersal) in island colonization, even if no correlation is found between species distribution and sea connectivity. Please, discuss with caution the topic of population connectivity, particularly when comparing the results of this study with the results of population genetic studies where gene flow gives a measure of the intensity of inter-island population connectivity. It should be clear in the Discussion that the species distribution data does not reflect connectivity per se, but rather at least one successful event of island colonization. This can be argued as the reason behind the absence of significant correlation between species distribution and sea connectivity.

The presentation of the statistical analysis' results in the manuscript should be improved, particularly of the (generalized) linear mixed models. The authors fitted LMM and GLMM sometimes using more than one predictor variable. However, they do not mention how well the models fit the data. What did the residuals of the models look like? Were there any violations of the model assumptions? Did any of the models experience convergence or singularity issues? Did adding a random factor improve the models? Given the small amount of data (only 14 species), I am aware that the models will not be perfect, and I suspect that the most complex ones might have problems. This should be shown and discussed in the manuscript. Thus, I recommend that the authors: (1) check the residuals of the models and mention this in the text; (2) compare each fitted models with simpler ones (first testing the random structure and then the fixed effects), and interpret the best model; (3) show the estimate for each predictor variable, not just the result of the statistical test, as the estimate specifies the strength and sign of the effect on the response. All of this will make the analyses more robust and reliable. There are quite a few papers and books explaining GLMMs and best practices; here, I include just two references the authors might find useful: Harrison et al (2008); Bolker et al. (2009).

Here, I detailed some suggestions focused on the different sections of the manuscript:

Abstract: It gives the reader a clear idea of the main question, methods and results. However, rephrasing the final part would make it more engaging. I would recommend splitting the long sentence that mixes results and conclusion about the drivers of plant distribution (L. 39-43), and separating the results (no relationship between geographic distance and species distribution; human usage) from the main conclusion ("challenge associated with identifying drivers of distribution ..."). I would emphasize this final part, as it is the main message derived from the association analysis. I would also shorten the final sentence on future studies to convey that, although there are still open questions, this study is the necessary first step in understanding the role of sea currents in the distribution of fleshy-fruited species.

Introduction: The structure of the Introduction can be better improved, in the current version, I get the impression that different topics are introduced without being clearly connected with each other (defaunation, islands, LDD).

Methods and results: I appreciate the effort the authors have made to improve the clarity of this part of the manuscript. However, I still find a bit confusing the last part of the Methods section. From line 253 to the end of the section, the authors describe different analyses one after the other in a way that might be confusing for the reader. I suggest introducing clearly the different analyses using a brief description of the motivation behind. Dividing the section into two, for example "dispersal potential and species connectivity calculation" and "statistical analysis" might benefit the reading of this section.

Regarding the authors' attempt to use GDMs, as commented in their response to reviewers, it is indeed necessary to include coordinates in the input as instructed in the manual. However, if the 'geo' option in the gdm() function is set to 'FALSE', geographic distances are not calculated from the provided coordinates. The authors should therefore run the function with fake coordinate columns and provide the geographic distance matrix together with other predictor variables (e.g. island characteristics; be careful if including sea connectivity in the same model as geographic distance because their high correlation), and set 'geo=FALSE'. This way it should run smoothly. Maybe it is not that interesting fitting GDM for each species independently, as there should be many NAs in most of them. However, it might be worthwhile implementing them with Sørensen matrices as the response (non-floating and floating community dissimilarity).

Minor comments:

L. 57 & 58: use the same type of dash to indicate the break in the sentence structure.

L. 64: the Rey & Alcántara (2000) ([11]) reference is about a single fleshy-fruited species (Olea europaea), and it is not sufficient on its own to support the authors' general statement about the importance of frugivorous seed LDD on isolated ecosystems. Please find additional papers that support this general statement.

L. 85 & 86: semicolons before the references should be removed. Also, replace the hyphens by either "or" or parentheses.

L. 85: the Costea et al. (2019) ([25]) study does indeed focus on endozoochory, but not driven by frugivorous birds. Instead, they demonstrate the important role of granivorous birds on plant dispersal. Please either find a more suitable reference or rephrase.

L. 86: the Harries & Clement (2014) ([27]) reference is about the coconut palm and it does not reference epizoochory.

L. 86: the extra period at the end of the sentence should be removed.

L. 90-92: from this sentence, I understand that the authors are talking about thalassochory in fleshy-fruited plants. However, the references [28-30] are either general references assessing different species [28 & 30] or focused on a single non-fleshy-fruited species [29]. Please rephrase to improve the clarity of the reasoning here.

L. 105: remove the comma before "(2)".

L. 114-125: Please include the number of islands in the study area. Knowing the total number of islands would give the reader a better idea of how well each species is distributed when looking at Table 1 data.

Table 1: Is 'initial count' the total number of fruits? If so, please specify. Also, be careful with the formatting. For example, the Chrysobalanus icaco row has a different font size and alignment.

Figure 2: What does it represent, the floatability or the viability of the seeds? Based on Table 1 data, I do not understand why Chrysobalanus icaco does not reach the 95th percentile line in the plot, and why some species do not reach zero. Please clarify it in the Figure caption.

L. 182-183: I guess the authors did some extrapolation combining data of the percentage of floating fruits obtained twice a week and the percentage of viable seeds obtained once a week for obtaining the curves showed in Fig. 2. More detail is needed to be able to follow how the authors did this calculation. Does the y-axis in Fig. 2 represent the percentage of floating fruits instead of viable floating fruits?

L. 190-192: Which is the response variable? Because I understand the question as the influence of fruit characteristics on the ability to float and survive in sea water. Thus, I would say that the response should be the maximum or the 95th percentile viability, and the predictors the fruit volume and density (check for collinearity). However, I understand the opposite from the text. Please clarify.

L. 254-259: Please clarify the terms of the formula. Is plant survival a percentage of viable floating seeds at a given time based on different 'Buoy time'? This 'Buoy time' is the transit time of each of the shared buoys between two islands? If so, why 'n' is not the number of shared buoys? Is connectivity the proportion of fruits that could successfully move between each island pair?

L. 267: What R package was used for fitting the GLMM?

L. 279: "each species' distribution matrix", please specify the number of species with which this analysis was performed. In Table 1, it is shown that some species are distributed only in 1-2 islands, and I presume that these were removed from the analysis.

Figure 3: Another plot can be included showing the effect of human usage in species distribution, as this predictor was also significant in the model.

L. 328: "or geographic distance", there is no reference to a correlation test between species distribution and geographic distance (L. 274-276). Same for L. 332 (L. 277-282 in Methods).

L. 328-330: If in L. 330-332 the authors explain the results of the correlation test between species similarities of non-floating and floating communities with metrics of ocean dispersal, what is this previous result? In methods is no other correlation test assessing the relationship of other community similarity matrix with dispersal metrics.

L. 366: Using geographic distance in biogeographic studies provides a reliable objective measure of isolation, and it makes sense that wind or oceanic currents are correlated with the geographic distance. Therefore, in my opinion, it seems somewhat pretentious to claim that "our results largely justify the usage of geographic distance in biogeographic studies".

L. 372-374: Do the data show evidence of this? I am thinking of species that, although they are theoretically able to reach one island from another by their capacity to float for the estimated minimum time required for colonization, are absent from the island and have a low probability of dispersing between the two islands. This would support the discussion in this paragraph; otherwise, I do not see the interest of this discussion, as no differences were found in the correlation analyses using Sørensen dissimilarity matrices with either dispersal metric. A single successful LDD event between islands might have been enough for a species to colonize and establish on an island.

L. 387: The fact that C. icaco possesses hollow seeds does not reduce the density of the whole fruit? How did the authors calculated fruit density? Using only the pulp? This should be clearly described in Methods.

L. 400: Please use with caution the term 'adapted' as it implies the action of selective forces in a certain direction. It should be noted that the ability to float and survive in sea water might not be the intended adaptation, but rather a consequence of trait evolution towards other processes/functionalities (e.g. dispersal by birds, protecting the seed, etc.). It is preferable to use 'better suited' instead.

L. 408: As commented before, it is not mention in Methods that geographic distance was used to evaluate its correlation with species distribution.

L. 409-410: A larger number of species must be used to claim this. With 14 species and most of them distributed in less than 10 islands out of 44, the similarity matrix used must not be representative of the real flora similarities between islands.

References:

Nathan, R. (2006). Long-distance dispersal of plants. Science, 313(5788), 786-788. DOI: 10.1126/science.1124975.

Nogales, M., Heleno, R., Traveset, A., & Vargas, P. (2012). Evidence for overlooked mechanisms of long-distance seed dispersal to and between oceanic islands. New Phytologist, 194(2), 313-317.

Vargas, P., Heleno, R., Traveset, A., & Nogales, M. (2012). Colonization of the Galápagos Islands by plants with no specific syndromes for long‐distance dispersal: a new perspective. Ecography, 35(1), 33-43. https://doi.org/10.1111/j.1600-0587.2011.06980.x

Esteves, C. F., Costa, J. M., Vargas, P., Freitas, H., & Heleno, R. H. (2015). On the limited potential of Azorean fleshy fruits for oceanic dispersal. PLoS One, 10(10), e0138882. https://doi.org/10.1371/journal.pone.0138882

Harrison, X. A., Donaldson, L., Correa-Cano, M. E., Evans, J., Fisher, D. N., Goodwin, C. E., ... & Inger, R. (2018). A brief introduction to mixed effects modelling and multi-model inference in ecology. PeerJ, 6, e4794.

Bolker, B. M., Brooks, M. E., Clark, C. J., Geange, S. W., Poulsen, J. R., Stevens, M. H. H., & White, J. S. S. (2009). Generalized linear mixed models: a practical guide for ecology and evolution. Trends in ecology & evolution, 24(3), 127-135.

Reviewer #3: I want to commend the authors on a very thorough revision that has greatly improved the study. I only have a few suggestions for a few further improvements (Note: All line numbers are for the version ‘Revised Manuscript with Track Changes’!):

Title: The new title is much better, but could potentially be improved even further (the current one puts major emphasis on the ocean current part of the study, which I find a bit of a shame, given the wonderful experimental flotation part of the study). Perhaps something like, “Testing the oceanic [or thalassochoric] dispersal potential of Carribean fleshy-fruited plants” – which would encompass both aspects into one (flotation + currents = oceanic dispersal).

I am a bit puzzled over the current styling of Fig 2 and Fig 3 and their titles in the main text – should these be on separate lines to show where the figures should be inserted? Especially the line/legend for Fig 2 (L422) is puzzling, as it contains a statistical result. Please correct/revise.

Depending on journal policy, it may not be possible for you to refer to the new co-author’s work as an ‘external database’, with the reference for this being an ‘in prep’ manuscript. Best to either cite the thesis (?) this comes from, or as ‘unpubl. data'. Check with the editor which is best.

L67 – insert comma after ‘context’.

L101 – correct to ‘specialized for’

L122 – suggest changing ‘possess’ to ‘have’

L365 – delete ‘While’ to start sentence with ‘The largest’

L497 – change to ‘our study’

L958 – correct to ‘Flotation’ in first word of legend.

Figure 2 – italicize the species names

7. PLOS authors have the option to publish the peer review history of their article (what does this mean?). If published, this will include your full peer review and any attached files.

Reviewer #1: No

Reviewer #2: **Yes:** Yurena Arjona

Reviewer #3: No

---

## [Author Response · Author response to Decision Letter 2]

3 Feb 2026

Seokmin Kim

Reviewer Responses

PLOS One

January 31, 2025

Reviewer #1: I feel the authors satisfactorily addressed the concerns raised by the reviews which contributed to a clearer and stronger manuscript. I therefore believe the manuscript is valid contribution to the field and I would like to congratulate the authors for their nice study. I have only a few specific points for consideration that I outline below.

Thank you for your comments and support. We appreciate you helping improve this manuscript ad hope that our new responses are satisfactory.

Specific comments

L85-86, I’m not sure if the 3 references cited in this sentence actually “demonstrate” that this mechanisms work for inter-island seed dispersal as neither of them managed to record the origin, vector and arrival of o colonizing propagule. We know that these 3 mechanisms are the potential ones because there are the only ones that can promote these movements, because they have been demonstrated to work on another contexts (not between islands), and because there is indirect probabilistic evidence suggesting that they are likely effective vectors. Please revise the term “demonstrated”, and also check that the references are the correct one (e.g. the paper dealing with the dispersal of the coconut palm is indicated as a demonstration of epizoochory). Also correct “..”

Changed “demonstrated” to “There are other mechanisms by which plants can disperse between islands such as anemochory – wind dispersal, [27] and epizoochory – attachment to feathers, fur, or skin [28].” (Line 86-87) Changed reference #28. Removed the extra period.

L96 The concept of “artificial introductions” is unclear. Consider replacing with anthropogenic introductions or a similar term.

Changed to “anthropogenic introductions”

L284 correct “..”

Corrected

L309 consider: visit -> visiting

We kept the original wording for this.

L435-443 I’m not fully convinced that “growth rate” is a particularly informative variable to include in future studies. Even if plants exposed to salty environments grow more slowly due to physiologic constraints, they can still promote gene flow and new colonization events (only slower). Moreover, as the current manuscript is not about growth rates, this seems like a bad choice of topic to close the manuscript.

The goal of our study is to determine whether thalassochory could replace endozoochory for plants that lack extant animals large enough to effectively disperse them. Thus, while it is true that as long as any viable seed can make it from one island to another, that thalassochory can contribute to the dispersal of that species, the degree to which thalassochory can replace endozoochory is affected by all other influences of saltwater exposure on seed fitness. Thus, we feel that growth rate after germination is a relevant demographic metric to track in future work.

Reviewer #2: The revised version of the manuscript, now entitled "Testing the potential of Caribbean fleshy-fruited plants to disperse using ocean currents" by Kim et al., has improved compared to the first version I read. I appreciate the effort made by the authors in addressing all of the reviewer's concerns. However, I believe further work is still needed on building the background to provide a clearer framework for the study, and on presenting the statistical results.

More care should be taken in relating the topics of the effect of defaunation on fleshy fruit dispersal and the long-distance dispersal (LDD) of plants. Long-distance dispersal is a rare process in nature compared to dispersal over shorter distances (long tail of the species' dispersal kernel). Different dispersal vectors may play a role in this process, and these vectors do not necessarily align with plant dispersal specializations (non-standard means of dispersal, see e.g. Nathan 2006; Nogales et al. 2012; Vargas et al. 2012). This decoupling between the plant dispersal syndrome (in this case, fleshy fruits, that are indicative of an endozoochorous syndrome) and the dispersal vector opens the door to investigating how well suited are plant diaspores to be dispersed by non-standard vectors (e.g. fleshy fruits dispersed by sea). However, throughout the text, the authors sometimes give the impression that they are referring to dispersal as a deterministic process (e.g. L. 341-342), and that plants only look for another appropriate vector when the standard vector is missing (e.g. sea dispersal as an alternative to endozoochory in the absence of frugivores). I find the defaunation phenomenon an interesting motivation to assess the ability of fleshy fruits to be dispersed by other means, like sea dispersal. However, stronger evidence of the effect of the decline of frugivores on LDD, especially on islands, is needed. As another reviewer pointed out, defaunation may pose a significant challenge to inland plant dispersal (over short-medium distances) but not to crossing the sea barrier between islands (LDD). In line with my suggestion about LDD and plant dispersal syndromes, the authors can use the introduction of Esteves et al (2015), where the authors assessed the potential of Azorean fleshy fruits to be dispersed by sea currents, as a reference.

Thank you for your detailed suggestions. Regarding the introduction, we have rearranged the beginning of it so that it now starts by discussing dispersal, connectivity, and LDD. We then explain why endozoochorous, fleshy-fruited plants are interesting within this context and elaborate on how these have been affected by defaunation. Next, we explore how fleshy-fruited plants could also utilize thalassochory as a LDD mechanism before going onto the rest of the introduction as before.

Another concern I have, especially after reading the Discussion, is the use and interpretation of 'population connectivity' in the framework of this study. This study is looking at the species presence/absence in the Caribbean islands, which must indeed be the result of some connection between islands. However, this does not mean that the connection is permanent or detectable nowadays. For example, an island could be colonized by a plant species after a single successful event of LDD followed by a complete isolation. Alternatively, a different regime of sea currents in the past might have connected two currently isolated islands. Presence/absence data cannot rule out the role of sea dispersal (or any other kind of dispersal) in island colonization, even if no correlation is found between species distribution and sea connectivity. Please, discuss with caution the topic of population connectivity, particularly when comparing the results of this study with the results of population genetic studies where gene flow gives a measure of the intensity of inter-island population connectivity. It should be clear in the Discussion that the species distribution data does not reflect connectivity per se, but rather at least one successful event of island colonization. This can be argued as the reason behind the absence of significant correlation between species distribution and sea connectivity.

We added the following in lines 429-433 “. It only takes a single successful LDD event to colonize a new island [5, 6]. The rare and stochastic nature of such events may make it difficult to correlate their outcome (species distributions across islands) with mean connectivity metrics, whether they be purely distance based or measured empirically based on ocean currents.” We hope that this clarifies our discussion with regards to your concerns.

The presentation of the statistical analysis' results in the manuscript should be improved, particularly of the (generalized) linear mixed models. The authors fitted LMM and GLMM sometimes using more than one predictor variable. However, they do not mention how well the models fit the data. What did the residuals of the models look like? Were there any violations of the model assumptions? Did any of the models experience convergence or singularity issues? Did adding a random factor improve the models? Given the small amount of data (only 14 species), I am aware that the models will not be perfect, and I suspect that the most complex ones might have problems. This should be shown and discussed in the manuscript. Thus, I recommend that the authors: (1) check the residuals of the models and mention this in the text; (2) compare each fitted models with simpler ones (first testing the random structure and then the fixed effects), and interpret the best model; (3) show the estimate for each predictor variable, not just the result of the statistical test, as the estimate specifies the strength and sign of the effect on the response. All of this will make the analyses more robust and reliable. There are quite a few papers and books explaining GLMMs and best practices; here, I include just two references the authors might find useful: Harrison et al (2008); Bolker et al. (2009).

In our revision, we checked residuals and convergence/singularity and found no issues. We also added estimates and their standard error into the text and state how there were no serious issues with model assumptions. See line 338-344: “(Fig 3: max: estimate (standard error) = 0.19 (0.086), z = 2.18, p = 0.03; 95th percentile: estimate (standard error) = 0.25 (0.031), z = 8.08, p < 0.001). This was accompanied by a significant, positive relationship between species range size and the sum of human use categories as well (Fig 3: estimate (standard error) = 0.22 (0.082), max: z = 2.72, p = 0.006; 95th percentile: estimate (standard error) = 0.25 (0.075), z = 3.36, p < 0.001). Residual plots against predicted values for both sets of models showed no systemic patterns, and none of the models experienced convergence or singularity issues.”

Regarding comparisons between fitted models and simpler ones, models with and without random effects were within a delta AIC of 2. Removing the random effect did not change the significance of any of the fixed effects. While the model without the random effect had a lower AIC, due to our concern about accounting for phylogenetic non-independence, we reported the model with the random effect even though the AIC was marginally higher.

Here, I detailed some suggestions focused on the different sections of the manuscript:

Abstract: It gives the reader a clear idea of the main question, methods and results. However, rephrasing the final part would make it more engaging. I would recommend splitting the long sentence that mixes results and conclusion about the drivers of plant distribution (L. 39-43), and separating the results (no relationship between geographic distance and species distribution; human usage) from the main conclusion ("challenge associated with identifying drivers of distribution ..."). I would emphasize this final part, as it is the main message derived from the association analysis. I would also shorten the final sentence on future studies to convey that, although there are still open questions, this study is the necessary first step in understanding the role of sea currents in the distribution of fleshy-fruited species.

We changed lines 39-46 to “Geographic distance, the isolation metric traditionally used in island biogeography studies, could not explain distribution or community patterns either, while human usage was identified as an alternative significant predictor of species range size. Our results thus illustrate the challenges associated with identifying drivers of distribution patterns across the Caribbean archipelagos. While we assert that future studies, such as those examining genetic connectivity of plant populations in the context of thalassochoric connectivity, are needed, this study serves as a crucial first step in understanding the role of sea currents in the distribution of fleshy-fruited plants.”

Introduction: The structure of the Introduction can be better improved, in the current version, I get the impression that different topics are introduced without being clearly connected with each other (defaunation, islands, LDD).

We have rearranged the beginning of it so that it now starts by discussing dispersal, connectivity, and LDD. We then explain why endozoochorous, fleshy-fruited plants are interesting within this context and elaborate on how these have been affected by defaunation. Next, we explore how fleshy-fruited plants could also utilize thalassochory as a LDD mechanism before going onto the rest of the introduction as before.

Methods and results: I appreciate the effort the authors have made to improve the clarity of this part of the manuscript. However, I still find a bit confusing the last part of the Methods section. From line 253 to the end of the section, the authors describe different analyses one after the other in a way that might be confusing for the reader. I suggest introducing clearly the different analyses using a brief description of the motivation behind. Dividing the section into two, for example "dispersal potential and species connectivity calculation" and "statistical analysis" might benefit the reading of this section.

We separated the methods into two sections as suggested.

Regarding the authors' attempt to use GDMs, as commented in their response to reviewers, it is indeed necessary to include coordinates in the input as instructed in the manual. However, if the 'geo' option in the gdm() function is set to 'FALSE', geographic distances are not calculated from the provided coordinates. The authors should therefore run the function with fake coordinate columns and provide the geographic distance matrix together with other predictor variables (e.g. island characteristics; be careful if including sea connectivity in the same model as geographic distance because their high correlation), and set 'geo=FALSE'. This way it should run smoothly. Maybe it is not that interesting fitting GDM for each species independently, as there should be many NAs in most of them. However, it might be worthwhile implementing them with Sørensen matrices as the response (non-floating and floating community dissimilarity).

Thank you for your suggestion. I did not consider providing fake coordinate columns. That allowed me to use the “formatsitepair” function, which I was previously unable to use. However, we are still getting errors in the final step despite putting geo = FALSE.

Error in gdm(sitePairTable, geo = FALSE) :

NA/NaN/Inf in foreign function call (arg 8)

In addition: Warning messages:

1: In mean.default(sort(x, partial = half + 0L:1L)[half + 0L:1L]) :

argument is not numeric or logical: returning NA

2: In gdm(sitePairTable, geo = FALSE) : NAs introduced by coercion

We get this despite making sure there aren’t NA values in our dataframes.

With some other dataframes, we also get this error. We then checked to make sure the matrices had the same number of rows and columns, but we still got the same error.

Error in createsitepair(dist = distData, spdata = bioData, envInfo = predData, :

The length of distance values is not the same as the expected number of rows in the site-pair table, unable to proceed.

Given the difficulties of running this model, we decided not to include GDMs in our manuscript.

Minor comments:

L. 57 & 58: use the same type of dash to indicate the break in the sentence structure.

Changed.

L. 64: the Rey & Alcántara (2000) ([11]) reference is about a single fleshy-fruited species (Olea europaea), and it is not sufficient on its own to support the authors' general statement about the importance of frugivorous seed LDD on isolated ecosystems. Please find additional papers that support this general statement.

Merged citations so that they are listed at the end of the sentence. Additionally, added Carlquist (1967) and Vargas et al. (2014) as references.

L. 85 & 86: semicolons before the references should be removed. Also, replace the hyphens by either "or" or parentheses.

Fixed

L. 85: the Costea et al. (2019) ([25]) study does indeed focus on endozoochory, but not driven by frugivorous birds. Instead, they demonstrate the important role of granivorous birds on plant dispersal. Please either find a more suitable reference or rephrase.

We changed this sentence in response to a re

---

## [Decision Letter · Decision Letter 2]

1 Apr 2026

PONE-D-25-10991R2Testing the oceanic dispersal potential of Caribbean fleshy-fruited plantsPLOS One

Dear Dr. Kim,

Thank you for submitting your manuscript to PLOS ONE. After careful consideration, we feel that it has merit but does not fully meet PLOS ONE’s publication criteria as it currently stands. Therefore, we invite you to submit a revised version of the manuscript that addresses the points raised during the review process.

I would like to acknowledge the considerable effort made by the authors in addressing all the reviewers’ comments. I believe that the revised version of the manuscript has improved substantially and will be a valuable and appreciated contribution for *PLOS ONE* readers. One of the reviewers has provided a few final minor suggestions; once these are addressed, I will be pleased to accept the manuscript.

We look forward to receiving your revised manuscript.

Kind regards,

Vicente Martínez López

Academic Editor

PLOS One

Journal Requirements:

Reviewers' comments:

Reviewer's Responses to Questions

**Comments to the Author**

1. If the authors have adequately addressed your comments raised in a previous round of review and you feel that this manuscript is now acceptable for publication, you may indicate that here to bypass the “Comments to the Author” section, enter your conflict of interest statement in the “Confidential to Editor” section, and submit your "Accept" recommendation.

Reviewer #1: All comments have been addressed

Reviewer #2: (No Response)

2. Is the manuscript technically sound, and do the data support the conclusions?

Reviewer #1: Yes

Reviewer #2: (No Response)

3. Has the statistical analysis been performed appropriately and rigorously? 

Reviewer #1: Yes

Reviewer #2: (No Response)

4. Have the authors made all data underlying the findings in their manuscript fully available?

Reviewer #1: Yes

Reviewer #2: (No Response)

5. Is the manuscript presented in an intelligible fashion and written in standard English?

Reviewer #1: Yes

Reviewer #2: Yes

6. Review Comments to the Author

Reviewer #1: This is my third review of this manuscript, and already on my previous review I identified no relevant concerns. The minor issued raised on my second review have now been properly solved, and I'm satisfied with the changes made.

Reviewer #2: The revised version of the manuscript, now entitled "Testing the oceanic dispersal potential of Caribbean fleshy-fruited plants" by Kim et al., has notably improved through the revision process. I appreciate the authors' effort in addressing the reviewer's concerns. This new version of the manuscript is clearer and better structured, and the use of the bibliography is also more coherent.

I only have some minor comments:

L. 68: It could be rephrased like "Another mechanism potentially involved in LDD of fleshy fruited plants" to avoid the use here of "utilize".

L. 87: Please take care of the use of punctuation here. The reference "[27]" should come before the comma as it refers to "wind dispersal". After the reference, the break is over, so mark it with another dash: "such as anemochory — wind dispersal [27] — and epizoochory — attachment to feathers".

L. 95: Do those large-fruited plants have always fleshy fruits? A clarification here might be good to link the previous and the next paragraphs.

L. 106-112: Include the assessment of the effect of human usage in the specific objectives and predictions.

L. 175: 10% of the floating fruits, right?

L. 193-194: Based on Table 1, the two response variables used here are counts (number of days), for this kind of data a generalized linear mixed model fitted with a Poisson family is more adequate.

L. 308: 6% "of the fruits"

L. 308-310: I would place this sentence at the end of the paragraph, after discussing all flotability and viability results per species.

L. 361: "Utilize" sounds a bit awkward in this context, as it evokes an intended decision.

L. 403: Should it not be "density and size" instead of "buoyancy"? Fruit density and size were the fruit characteristics assessed here in relation to species flotability.

L. 508: The doi corresponds to the supplementary material of the paper.

7. PLOS authors have the option to publish the peer review history of their article (what does this mean?). If published, this will include your full peer review and any attached files.

Reviewer #1: No

Reviewer #2: **Yes:** Yurena Arjona

---

## [Author Response · Author response to Decision Letter 3]

15 Apr 2026

Reviewer #1: This is my third review of this manuscript, and already on my previous review I identified no relevant concerns. The minor issued raised on my second review have now been properly solved, and I'm satisfied with the changes made.

Thank you for your comments and support. We appreciate you helping improve this manuscript.

Reviewer #2: The revised version of the manuscript, now entitled "Testing the oceanic dispersal potential of Caribbean fleshy-fruited plants" by Kim et al., has notably improved through the revision process. I appreciate the authors' effort in addressing the reviewer's concerns. This new version of the manuscript is clearer and better structured, and the use of the bibliography is also more coherent.

Thank you for your comments and support. We appreciate you helping improve this manuscript.

I only have some minor comments:

L. 68: It could be rephrased like "Another mechanism potentially involved in LDD of fleshy fruited plants" to avoid the use here of "utilize".

Change made.

L. 87: Please take care of the use of punctuation here. The reference "[27]" should come before the comma as it refers to "wind dispersal". After the reference, the break is over, so mark it with another dash: "such as anemochory — wind dispersal [27] — and epizoochory — attachment to feathers".

Change made.

L. 95: Do those large-fruited plants have always fleshy fruits? A clarification here might be good to link the previous and the next paragraphs.

Clarified to “large, fleshy-fruited plants”

L. 106-112: Include the assessment of the effect of human usage in the specific objectives and predictions.

Added “(4) Can human usage serve as an alternative predictor of species distributions?” in line 108-109.

L. 175: 10% of the floating fruits, right?

Correct. Now reads: “To conduct the tetrazolium test, we arbitrarily removed 10% of the floating fruits per species…”

L. 193-194: Based on Table 1, the two response variables used here are counts (number of days), for this kind of data a generalized linear mixed model fitted with a Poisson family is more adequate.

Clarified it to: “We then conducted generalized linear mixed model regressions fitted with a Poisson family…”

L. 308: 6% "of the fruits"

Corrected.

L. 308-310: I would place this sentence at the end of the paragraph, after discussing all flotability and viability results per species.

Moved.

L. 361: "Utilize" sounds a bit awkward in this context, as it evokes an intended decision.

Changed to “these plants can also benefit from”

L. 403: Should it not be "density and size" instead of "buoyancy"? Fruit density and size were the fruit characteristics assessed here in relation to species flotability.

We think buoyancy is more appropriate here as that was what we were ultimately aiming to evaluate. We mention density and size as the characteristics that we assessed in the previous sentence.

L. 508: The doi corresponds to the supplementary material of the paper.

Fixed.

---

## [Editor Report · Decision Letter 3]

19 Apr 2026

Testing the oceanic dispersal potential of Caribbean fleshy-fruited plants

PONE-D-25-10991R3

Dear Dr. Kim,

We’re pleased to inform you that your manuscript has been judged scientifically suitable for publication and will be formally accepted for publication once it meets all outstanding technical requirements.

Kind regards,

Vicente Martínez López

Academic Editor

PLOS One
---

## [Editor Report · Acceptance letter]

PONE-D-25-10991R3

PLOS One

Dear Dr. Kim,

I'm pleased to inform you that your manuscript has been deemed suitable for publication in PLOS One. Congratulations! Your manuscript is now being handed over to our production team.

Kind regards,

on behalf of

Dr. Vicente Martínez López

Academic Editor

PLOS One